# Turning Back Without Forgetting: Selective Backward Refinement for Parameter-Efficient Continual Learning

Anushka Tiwari [1]  Kaiyi Ji [2]

## Abstract

While prompt-based parameter-efficient continual learning mitigates catastrophic forgetting by isolating task-specific prompts, this isolation also limits later tasks from improving earlier ones, leaving backward knowledge transfer underexplored. We address this limitation by proposing **S**elective b**A**ckward refinement for positive **B**ackward knowledge transf**ER** (SABER), a replay-free framework that enables controlled backward transfer in prompt-based continual learning. SABER determines *when* backward refinement is beneficial using complementary task-correlation criteria based on prompt-gradient geometry and loss-distribution similarity, and *how* to perform refinement safely by restricting updates to non-interfering directions in the prompt parameter space. Extensive experiments across multiple continual learning benchmarks, and diverse pretrained backbones, including T5-Large, LLaMA, and Qwen, demonstrate that SABER consistently achieves positive backward transfer while maintaining strong overall average performance. Code is available at https://github.com/OptMN-Lab/SABER-ICML-2026/.

## 1. Introduction

Continual learning (CL) (Van de Ven & Tolias, 2018) studies how models can be trained on a sequence of tasks over time while maintaining strong performance on earlier tasks. In recent years, parameter-efficient fine tuning (PEFT) (Wang et al., 2022c; Smith et al., 2023; Feng et al., 2024) has emerged as a central paradigm in continual learning, offering a computationally efficient alternative to full model retraining for large pretrained models. By restricting up-

dates to a small set of task-specific parameters, such as prompts (Razdaibiedina et al., 2023; Feng et al., 2024; Tiwari et al., 2025b), adapters (Yu et al., 2024; Zhao et al., 2024), or low-rank weight updates (Liang & Li, 2024; Wu et al., 2025), PEFT methods have proven effective at mitigating catastrophic forgetting.

Among PEFT-based continual learning approaches, prompt-based methods have emerged as an effective and lightweight solution. Rather than updating backbone weights or introducing additional trainable modules, these methods represent task-specific knowledge through soft prompts that are prepended to the input (or injected into intermediate representations), while keeping the pretrained model entirely frozen (Lester et al., 2021b; Li & Liang, 2021). As a result, prompt-based continual learning typically requires substantially fewer trainable parameters per task than adapter- or LoRA-based alternatives, making it particularly appealing for long task sequences and large-scale pretrained models. Building on this paradigm, a growing body of work has explored prompt-based continual learning via progressive prompt construction, prompt pools, and shared prompt mechanisms, aiming to maintain task isolation while enabling forward transfer across tasks (Wang et al., 2022c;b; Razdaibiedina et al., 2023; Smith et al., 2023; Zhao et al., 2024; Wu et al., 2024).

Despite their strong empirical performance, most prompt-based continual learning methods mitigate catastrophic forgetting by enforcing strict task isolation: once a task-specific prompt is learned, it is frozen and never updated as new tasks arrive (Wang et al., 2022b; Razdaibiedina et al., 2023; Wu et al., 2024). While effective for preserving past-task accuracy, this design also prevents later tasks from improving earlier task representations, even when tasks are related and shared knowledge could be beneficial (Lin et al., 2022; Chaudhry et al., 2018b). Recent work has started to explore backward knowledge transfer under parameter isolation (Wong et al., 2024; Li et al., 2026); for example, Wong et al. (2024) selectively updates task masks using gradient-based signals and replayed data to improve past-task performance, and Li et al. (2026) proposes causal-aware LoRA to guide current adapter updates using prior-task signals. However, these approaches do not directly refine task-specific

[1]Institute for Artificial Intelligence and Data Science, University at Buffalo, Buffalo, NY, USA [2]Department of Computer Science and Engineering, University at Buffalo, Buffalo, NY, USA. Correspondence to: Anushka Tiwari <atiwari4@buffalo.edu>.

*Proceedings of the $43^{rd}$ International Conference on Machine Learning*, Seoul, South Korea. PMLR 306, 2026. Copyright 2026 by the author(s).

*Table 1.* Comparison of continual prompt tuning methods across key properties. ✓ denotes satisfied properties; ✗ denotes otherwise.

| Method | Replay-Free | Update Prior Prompts | Positive Backward Transfer |
|---|---|---|---|
| LFPT5 (ICLR'22) | ✗ | ✗ | ✗ |
| ProgPrompt (ICLR'23) | ✓ | ✗ | ✗ |
| CODA-Prompt (CVPR'23) | ✓ | ✗ | ✗ |
| SHLPT (ACL'24) | ✓ | ✗ | ✗ |
| SAPT (ACL'24) | ✓ | ✗ | ✗ |
| **SABER (ours)** | ✓ | ✓ | ✓ |

representations once learned. In prompt-based continual learning, where task knowledge is primarily encoded in the prompts, freezing prior prompts therefore makes backward knowledge transfer inherently difficult, and leveraging later tasks to refine earlier prompts remains relatively underexplored (Guo et al., 2024; Zhao et al., 2024).

These observations motivate the design of prompt-based continual learning methods that not only preserve past knowledge, but also enable *backward* knowledge transfer to improve earlier tasks. To achieve this, two questions are critical. First, we need to decide *when* backward updates are helpful, since only sufficiently related tasks are likely to benefit from such transfer (Lin et al., 2022). Second, even when such relationships exist, the model must decide *how* to update task-specific parameters to promote positive transfer while avoiding negative interference (Saha et al., 2021). This calls for mechanisms that can identify task correlations and selectively regulate backward updates to balance performance improvement with knowledge preservation.

Our main contributions are summarized as follows:

- We propose **S**elective b**A**ckward refinement for positive **B**ackward knowledge transf**ER** (SABER), the first framework to enable a replay-free positive backward transfer in prompt-based continual learning. SABER involves (i) a task-correlation criterion to identify when backward refinement is beneficial and (ii) a selective backward prompt refinement mechanism that constrains updates to non-interfering directions, preserving previously learned knowledge. Our theoretical guarantees show that backward refinements are non-interfering and yield non-increasing loss under mild conditions.

- We evaluate SABER on multiple continual learning benchmarks with different task orders and diverse model backbones, including T5-Large, Qwen, and LLaMA, and observe consistent improvements in both average performance (AP) and backward knowledge transfer (BWT). Among the compared methods, SABER is the only approach that consistently exhibits positive backward knowledge transfer. We also show that SABER is modular and can be readily integrated into existing prompt-based continual learning frameworks, including frozen prompt pools and shared prompt augmentation.

**Conflict of Interest Disclosure.** The authors declare no financial conflicts of interest related to this work.

## 2. Related Work

**Continual learning.** Continual learning studies how models learn from a sequence of tasks without access to all past data, while mitigating catastrophic forgetting, where learning new tasks degrades performance on earlier ones (Mc-Closkey & Cohen, 1989). Existing CL methods are typically grouped into three categories: *memory-based* methods that replay past data (Shin et al., 2017; Bang et al., 2021; Hao et al., 2023), *regularization-based* methods that constrain parameter updates to preserve prior knowledge (Kirkpatrick et al., 2017; Zenke et al., 2017), and *architecture-based* methods that introduce task-specific components to isolate representations (Rusu et al., 2016; Lee et al., 2017; Wang et al., 2022b). While effective, these approaches face scalability challenges for large pretrained models, motivating parameter-efficient CL methods that rely on lightweight components such as prompts or adapters to enable continual learning with minimal parameter growth (Xu et al., 2023; Rücklé et al., 2021).

**Parameter-efficient continual learning.** PEFT adapts large pretrained models by adding lightweight, task-specific components while keeping the backbone frozen, enabling continual learning with low computational and memory cost. Typical approaches include adapters and low-rank adaptation, which reduce task interference by updating only a small number of additional parameters (Rücklé et al., 2021; Hu et al., 2022). Prompt tuning is a widely used PEFT strategy that learns task-specific continuous prompts appended to the input or intermediate representations (Lester et al., 2021a; Li & Liang, 2021; Gu et al., 2022). Continual prompt tuning extends this idea to task sequences via prompt concatenation, prompt pools, or shared prompts (Zhu et al., 2022; Yin et al., 2022; Wang et al., 2022b; Razdaibiedina et al., 2023; Smith et al., 2023). While effective at preserving prior performance, many existing methods either rely on replay buffers (Zhu et al., 2022) or avoid updating previously learned task components, which may hinder consistent positive backward transfer (Zhao et al., 2024; Wu et al., 2024; Qiao et al., 2024; Tiwari et al., 2025a; Lu et al., 2025).

**Task similarity and knowledge transfer.** Existing approaches leverage task similarity by expanding model architectures or shaping gradient updates, such as through gradient alignment or projection into task-specific subspaces to reduce interference (Rusu et al., 2016; Lee et al., 2017; Lopez-Paz & Ranzato, 2017; Saha et al., 2021; Farajtabar et al., 2020). These methods mainly use task similarity to guide forward learning or constrain updates, rather than to improve previously learned tasks. Recent work explores backward knowledge transfer using optimization-, architecture-, or replay-based strategies (Lin et al., 2022; Wang et al., 2026; Shin et al., 2017; Li et al., 2026), but typically relies on data replay or global updates. In contrast, we use task similarity to selectively refine prompts, enabling replay-free positive

backward transfer.

## 3. Problem Formulation

### 3.1. Problem Definition

Consider a task-incremental continual learning setting with a sequence of tasks $\mathcal{T} = \{T_1, \ldots, T_T\}$ arriving sequentially. Each task $T_t$ is associated with a labeled dataset $\mathcal{D}_t = \{(x_{t,i}, y_{t,i})\}_{i=1}^{N_t}$, where $x_{t,i}$ denotes the input and $y_{t,i}$ the corresponding label. We consider a pretrained language model $f(\cdot; \theta)$ with frozen parameters $\theta$, and adapt it to each task using soft prompts.

For each task $T_t$, a task-specific soft prompt $u_t \in \mathbb{R}^{\ell \times d}$ is defined, where $\ell$ denotes the prompt length and $d$ is the embedding dimension. After observing tasks $\{T_1, \ldots, T_{t-1}\}$, the set of learned prompts is maintained as $\mathcal{U}_{t-1} = \{u_1, \ldots, u_{t-1}\}$. When a new task $T_t$ arrives, a new prompt $u_t$ is initialized and optimized on $\mathcal{D}_t$ with the backbone $f(\cdot; \theta)$ fixed; optionally, a subset of previously learned prompts $\mathcal{S}_t \subseteq \mathcal{U}_{t-1}$ could be further updated. Let $\mathcal{U}^{(t)} = \mathcal{U}_{t-1} \cup \{u_t\}$ denote the prompt set at time $t$. Given an input $x$ and a prompt $u$, the model prediction is $\hat{y} = f(x; u)$, and the task-$t$ loss is $\mathcal{L}_t(u_t) = \mathbb{E}_{(x,y) \sim \mathcal{D}_t}\big[\ell\big(f(x; u_t), y\big)\big]$, where $\ell(\cdot)$ denotes the per-example loss. We further define $\mathcal{L}_t(B, u_t)$ as the mini-batch averaged loss on a batch $B \subset \mathcal{D}_t$ under prompt $u_t$.

### 3.2. Challenges of Positive Backward Transfer in Prompt-Based Continual Learning

Prompt-based CL addresses forgetting by isolating task-specific knowledge in lightweight prompts, but this isolation can hinder backward transfer: updating a previously learned prompt using gradients from a new task may introduce task-mismatched modifications and overwrite directions that were critical for the earlier task, resulting in negative backward transfer. *The following examples illustrate that unconstrained backward updates can be unreliable.*

**Example 1: Unrelated Tasks.** Consider two tasks $\mathcal{T}_1$ and $\mathcal{T}_2$ with distinct objectives and non-overlapping semantics, such as tasks with different input-output formats (e.g., entailment vs. sentiment analysis). Let $u_1$ denote the prompt learned for $\mathcal{T}_1$. During training on $\mathcal{T}_2$, performing unconstrained backward updates on $u_1$ using gradients from $\mathcal{T}_2$ can drive $u_1$ in directions that are misaligned with what is important for $\mathcal{T}_1$, thereby degrading performance on the earlier task. Table 2 demonstrates this phenomenon across several unrelated task pairs, where backward prompt updates consistently result in negative pairwise backward transfer.

**Example 2: Related Tasks.** Now consider tasks that are more closely related, e.g., those sharing similar input-output structures or underlying semantics. In this case, one might expect knowledge from $\mathcal{T}_2$ to refine the earlier prompt $u_1$

*Table 2.* Unconstrained backward prompt updates on **unrelated task pairs** lead to negative backward transfer. We report accuracy on the earlier task $T_1$ before and after updating its prompt during training on $T_2$. $\Delta$Acc denotes pairwise backward transfer (BWT).

| **Pair** $(T_1 \leftarrow T_2)$ | $\text{Acc}_{T_1}$ (Before) | $\text{Acc}_{T_1}$ (After) | $\Delta$Acc (BWT) |
|---|---|---|---|
| MNLI $\leftarrow$ Yahoo | 0.867 | 0.847 | -0.020 |
| QQP $\leftarrow$ Yelp | 0.484 | 0.472 | -0.012 |
| WiC $\leftarrow$ MultiRC | 0.514 | 0.354 | -0.160 |
| QQP $\leftarrow$ MultiRC | 0.476 | 0.454 | -0.022 |

and yield positive backward transfer. However, even when tasks are related, unconstrained backward updates can still disrupt directions in $u_1$ that are crucial for $\mathcal{T}_1$, which can diminish or eliminate any potential gains. Table 3 shows that such updates often result in negligible or even negative backward transfer, even for related task pairs.

*Table 3.* Even for **related task pairs**, unconstrained backward prompt updates fail to yield positive backward transfer. This motivates the need for geometry-aware update directions.

| **Pair** $(T_1 \leftarrow T_2)$ | $\text{Acc}_{T_1}$ (Before) | $\text{Acc}_{T_1}$ (After) | $\Delta$Acc (BWT) |
|---|---|---|---|
| IMDb $\leftarrow$ Amazon | 0.958 | 0.914 | -0.044 |
| Yelp $\leftarrow$ Amazon | 0.603 | 0.482 | -0.121 |
| IMDb $\leftarrow$ SST-2 | 0.950 | 0.948 | -0.002 |
| BoolQ $\leftarrow$ MultiRC | 0.832 | 0.792 | -0.040 |

Together, these examples highlight two requirements for achieving positive backward transfer in prompt-based continual learning. First, backward refinement needs to be *selective*: only a subset of prior tasks are compatible with the current task's learning signal, and semantic similarity or task labels alone do not reliably identify them. Second, backward refinement needs to be *constrained*: even for compatible tasks, unconstrained updates can still overwrite directions that are important for earlier prompts, limiting potential gains. These observations motivate principled criteria for task compatibility and geometry-aware update rules that encourage transfer while reducing interference.

## 4. Method

We build on the insights from Section 3.2 that backward refinement in continual learning is *not universally beneficial*, but depends critically on whether the current task's learning signal is compatible with previously learned tasks. This calls for a principled mechanism that can explicitly identify which prior prompts can benefit from refinement and enforce update rules that prevent interference. We therefore propose a framework for *selective backward refinement* that jointly determines *when* backward updates should be performed and *how* to apply them safely. An overview of the proposed framework is shown in Figure 1 in Appendix.

### 4.1. Task Correlation for Selective Backward Refinement

We assess task relatedness from two complementary perspectives: an *update-level* view, which characterizes compat-

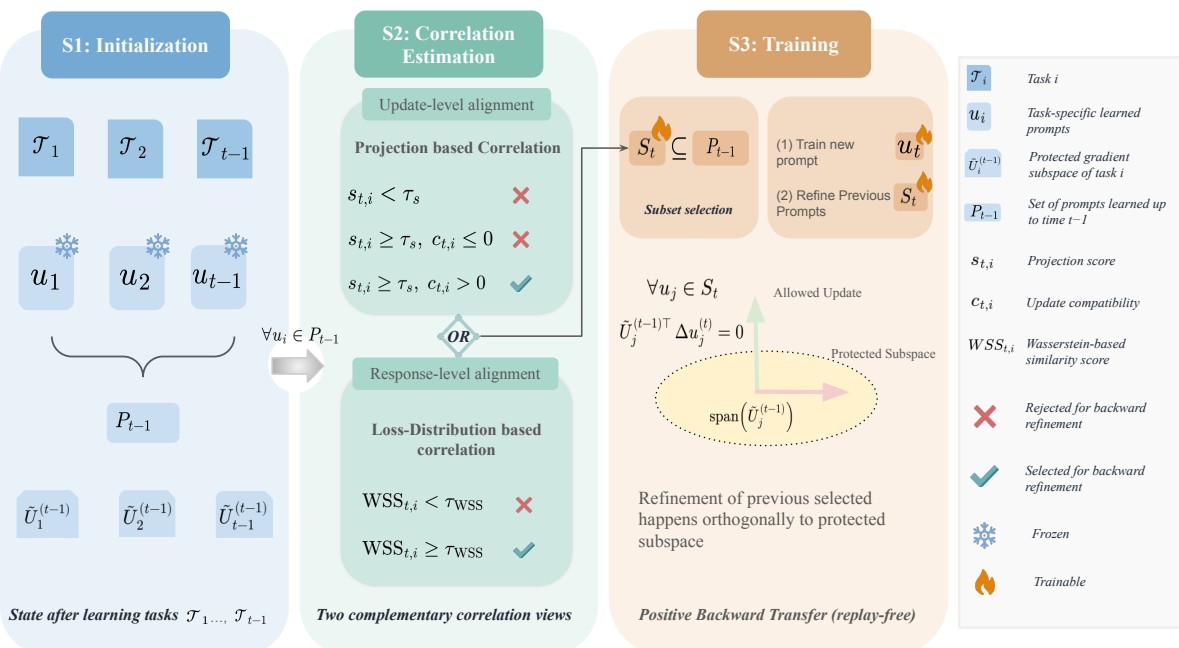

*Figure 1.* Overview of the proposed SABER framework. (S1) the state after learning tasks $\mathcal{T}_1, \ldots, \mathcal{T}_{t-1}$, where task-specific prompts have been learned and frozen and their corresponding protected gradient subspaces are maintained; at this point, the model is now at task $\mathcal{T}_t$ and a new prompt $u_t$ is initialized for learning; (S2) task correlation estimation using either update-level criteria (projection and compatibility) or response-level criteria (loss-distribution) to select prior prompts; and (S3) training on task $\mathcal{T}_t$, where $u_t$ is learned, and selected prior prompts are refined orthogonally to their protected subspaces, enabling positive backward transfer.

ibility through the geometry of parameter updates induced by the current task, and a *response-level* view, which compares how tasks induce similar model behavior as reflected in their loss responses.

### 4.1.1. PROJECTION-BASED TASK CORRELATION

For a prior task $\mathcal{T}_i$, we denote its task-specific prompt by $u_i \in \mathbb{R}^{\ell \times d}$. We collect prompt gradients $\nabla_{u_i} \mathcal{L}_i(u_i)$ over $\mathcal{D}_i$, and let $U_i \in \mathbb{R}^{(\ell d) \times r}$ be an orthonormal basis of the gradient subspace of the task $\mathcal{T}_i$, where $r \ll \ell d$ (see Appendix §B for details). We further denote the average gradient during training of $\mathcal{T}_i$ by $\bar{g}_i = \mathbb{E}_{B \sim \mathcal{D}_i}[\nabla_{u_i} \mathcal{L}_i(B; u_i)]$. For a new task $\mathcal{T}_t$, the gradient of the current-task loss with respect to $u_i$ is denoted as $g_{t \to i} = \nabla_{u_i} \mathcal{L}_t(u_i)$, with mean denoted by $\bar{g}_{t \to i} = \mathbb{E}_{B \sim \mathcal{D}_t}[\nabla_{u_i} \mathcal{L}_t(B; u_i)]$.

Task correlation is captured by projecting the current-task gradient $g_{t \to i}$ onto the subspace spanned by $U_i$ and computing the relative projection magnitude. Formally, the projection score is defined as

$$s_i = \mathbb{E}\left[\frac{\|U_i U_i^\top g_{t \to i}\|_2}{\|g_{t \to i}\|_2}\right]. \tag{1}$$

The score $s_i \in [0, 1]$ measures the fraction of the current-task update that lies within the gradient subspace identified during the training of task $\mathcal{T}_i$. A large value of $s_i$ suggests that $\mathcal{T}_i$ and $\mathcal{T}_t$ share sufficient common bases in the

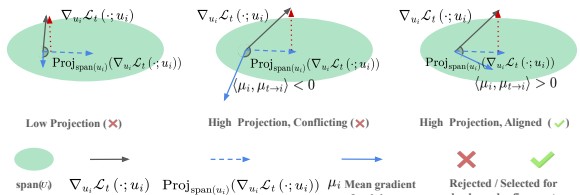

*Figure 2.* Projection-based task correlation used to select prior prompts for backward refinement. A prior task is selected only when the current-task gradient has sufficient projection onto its gradient subspace and is directionally aligned with its gradient.

prompt-parameter space, indicating stronger alignment at the update level. While a sufficient projection score suggests a potentially strong correlation, it does not ensure positive correlation.

We therefore quantify the *gradient alignment* between the current task $\mathcal{T}_t$ and the prior task $\mathcal{T}_i$ with respect to the prompt $u_i$ as:

$$c_i = \max\left(\frac{\langle \bar{g}_i, \bar{g}_{t \to i} \rangle}{\|\bar{g}_i\|_2 \|\bar{g}_{t \to i}\|_2}, 0\right). \tag{2}$$

A sufficient projection score $s_i$ together with $c_i > 0$ indicates positive correlation between tasks, and the potential for positive backward transfer from $\mathcal{T}_t$ to $\mathcal{T}_i$.

### 4.1.2. LOSS-DISTRIBUTION-BASED TASK CORRELATION

While projection-based task correlation captures compatibility between tasks at the level of prompt gradients and reflects local learning signals, task relatedness may also be assessed from a response-level perspective. In particular, distributions of per-batch losses summarize how the model responds to a task across inputs, providing a complementary and more global view of task similarity.

For each task $\mathcal{T}_i$, let $\ell(B; u)$ denote the mini-batch averaged loss evaluated on a batch $B$, where $u = \varnothing$ indicates that no prompt is used. For any task $\mathcal{T}_t$ and any prompt $u_i$ (including $u_i = \varnothing$), we define the empirical loss distribution $\mathcal{P}_t^{(i)} = \{\ell(B_j; u_i)\}_{j=1}^{|B_t|}$ with batches $B_j \sim \mathcal{D}_t$. Under this notation, $\mathcal{P}_i^{(0)}$ corresponds to the frozen-backbone loss distribution for task $\mathcal{T}_i$, and $\mathcal{P}_i^{(i)}$ denotes the prompt-induced loss distribution after learning the task-specific prompt $u_i$. When a new task $\mathcal{T}_t$ arrives, we compute $\mathcal{P}_t^{(0)}$ and, for each prior prompt $u_i$, the cross-task distribution $\mathcal{P}_t^{(i)}$. Finally, let $W(\cdot, \cdot)$ denote the Wasserstein distance between empirical loss distributions (Vallender, 1974; Oord et al., 2018).

To find correlation of task $\mathcal{T}_t$ with a prior task $\mathcal{T}_i$, define the backbone-level and prompt-induced distances as

$$d_i^{(0)} = W\left(\mathcal{P}_i^{(0)}, \mathcal{P}_t^{(0)}\right) \quad \text{and} \quad d_i^{(i)} = W\left(\mathcal{P}_i^{(i)}, \mathcal{P}_t^{(i)}\right).$$

The Wasserstein-based similarity score is then defined as

$$\text{WSS}_i = d_i^{(0)} - d_i^{(i)}. \tag{3}$$

The underlying intuition (see Figure 7 in the appendix) is that if the task-specific knowledge learned for $\mathcal{T}_i$ transfers to $\mathcal{T}_t$, then applying the prompt $u_i$ to both tasks is expected to reduce the discrepancy between their loss responses. Accordingly, $\text{WSS}_i > 0$ suggests that $u_i$ tends to bring the loss distributions of $\mathcal{T}_i$ and $\mathcal{T}_t$ closer relative to the frozen backbone, whereas little or no reduction may indicate weak response-level similarity.

*Comparison between projection-based and loss-distribution-based selections.* Both selection criteria are effective, but they offer complementary advantages. The projection-based criterion leverages prompt-gradient geometry to measure task correlation, capturing fine-grained update compatibility and enabling principled control of backward refinement (e.g., improved safety and stability). This geometric view is particularly useful when reliable gradient information is available and stronger guarantees are desired.

In contrast, the loss-distribution-based criterion operates directly at the response level and relies only on scalar loss statistics, making it substantially lighter to store and maintain. This simplicity becomes especially important in large-scale continual learning, where retaining task-specific gradient subspaces may incur nontrivial memory overhead as the number of tasks or the prompt dimensionality grows. Together, these two criteria provide a practical trade-off between granularity and efficiency, and we find that each can be preferable depending on the deployment constraints.

### 4.2. Constrained Backward Prompt Updates

Having identified prior prompts that are likely to yield *positive backward transfer*, we next consider how to update them safely. The main challenge is to enable backward refinement using gradients from the current task, while avoiding changes along directions that were essential for the earlier task. For a selected prior prompt $u_i$, let $U_i \in \mathbb{R}^{(\ell d) \times r}$ denote an orthonormal basis spanning the task-$i$ gradient subspace identified during training on $\mathcal{T}_i$. Since this subspace captures directions that are most influential for $\mathcal{T}_i$, we treat it as a *protected subspace*. Given the gradient of the current task $\mathcal{T}_t$ with respect to $u_i$, $g_{t \to i} = \nabla_{u_i} \mathcal{L}_t(u_i)$, we constrain backward refinement to the orthogonal complement of the protected subspace by projecting out the component aligned with $U_i$: $\Delta u_i^{\text{orth}} = (I - U_i U_i^\top) g_{t \to i}$. We then perform backward refinement via $u_i \leftarrow u_i - \eta \, \Delta u_i^{\text{orth}}$. This update exploits unused capacity in the prompt parameter space while largely preserving directions that are important for $\mathcal{T}_i$, reducing the risk of negative backward transfer.

*Rationality and comparison to other options.* As illustrated in Figure 3, we compare our orthogonal backward refinement with several alternative update strategies for a selected prior prompt $u_i$. Specifically, given the current-task gradient $g_{t \to i}$, we consider: (i) the unconstrained update $\Delta u_i^{\text{unconstr}} = g_{t \to i}$, (ii) the same-subspace update $\Delta u_i^{\text{same}} = U_i U_i^\top g_{t \to i}$, and (iii) a hybrid interpolation $\Delta u_i^{\text{hybrid}} = \alpha \Delta u_i^{\text{same}} + (1 - \alpha) \Delta u_i^{\text{orth}}$ with $\alpha \in (0, 1)$. A natural baseline is to directly apply current-task gradients without restriction. However, Table 4 shows that $\Delta u_i^{\text{unconstr}}$ often yields noisy and inconsistent backward transfer, suggesting substantial interference with task-$i$ knowledge. Restricting updates to the protected subspace, i.e., using $\Delta u_i^{\text{same}}$, typically performs even worse, indicating that modifying directions that were critical for $\mathcal{T}_i$ can be particularly harmful. While these directions encode prior-task information, uniformly perturbing them can overwrite finely tuned representations established during initial training. Hybrid updates inherit the same issue, and remain unstable across tasks.

Taken together, these results motivate $\Delta u_i^{\text{orth}}$ as a **simple yet effective mechanism** for backward refinement that minimizes interference with protected task-$i$ directions.

**Cumulative protected subspace.** While the above formulation describes a single backward refinement step, a prior prompt may be refined multiple times as new tasks arrive. To prevent successive refinements from revisiting or over-

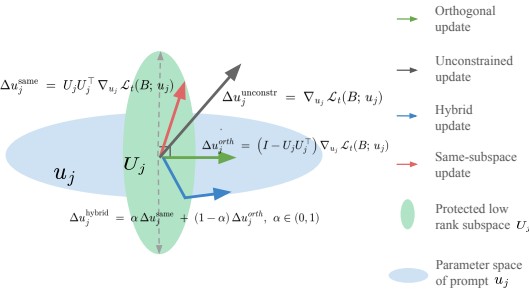

*Figure 3.* Geometry of different update strategies.

*Table 4.* Effect of backward update subspace choice on pairwise backward transfer. $\Delta$Acc denotes the change in accuracy on the earlier task after backward update during training on the later task.

| Pair $(T_i \leftarrow T_t)$ | Unconstr. | Same-sub. | Orthog. | Hybrid |
|---|---|---|---|---|
| MNLI ← CB | +0.001 | −0.003 | +0.030 | +0.000 |
| QQP ← MultiRC | +0.006 | +0.000 | +0.008 | +0.006 |
| IMDb ← Amazon | +0.010 | −0.002 | +0.020 | +0.008 |
| Yelp ← Amazon | +0.005 | −0.028 | +0.001 | −0.017 |
| BoolQ ← MultiRC | −0.040 | −0.050 | −0.010 | −0.048 |
| COPA ← BoolQ | +0.060 | +0.040 | +0.080 | +0.040 |
| **Avg. $\Delta$Acc** | +0.007 | −0.007 | +0.0215 | −0.002 |

writing directions explored earlier, we maintain a cumulative protected subspace, denoted as $\tilde{U}_i^{(t)} \in \mathbb{R}^{(\ell d) \times r_i^{(t)}}$, for each prompt $u_i$, which is initialized as $\tilde{U}_i^{(i)} \leftarrow U_i$. At time $t$, we define the *safe backward update* $\Delta u_i^{(t)}$ as the component of the current-task gradient that lies outside $\tilde{U}_i^{(t)}$,

$$\Delta u_i^{(t)} = \big(I - \tilde{U}_i^{(t-1)}\tilde{U}_i^{(t-1)\top}\big)\nabla_{u_i}\mathcal{L}_t(u_i). \qquad (4)$$

To ensure that newly explored directions are also protected from *future* interference, we augment the protected subspace after each backward refinement step. Specifically, we first normalize the safe update direction as $\hat{v}_i^{(t)} = \Delta u_i^{(t)}/\|\Delta u_i^{(t)}\|_2$, and explicitly remove any residual component in the current protected subspace $\mathrm{span}(\tilde{U}_i^{(t-1)})$:

$$v_i^{(t)} = \hat{v}_i^{(t)} - \tilde{U}_i^{(t-1)}\tilde{U}_i^{(t-1)\top}\hat{v}_i^{(t)}. \qquad (5)$$

If $v_i^{(t)} \neq 0$, we then append it to the protected subspace via orthonormalization,

$$\tilde{U}_i^{(t)} \leftarrow \mathrm{orth}\Big(\big[\tilde{U}_i^{(t-1)},\ v_i^{(t)}/\|v_i^{(t)}\|_2\big]\Big), \qquad (6)$$

where $\mathrm{orth}(\cdot)$ denotes an orthonormalization operator that returns an orthonormal basis spanning the same column space. As a result, $\tilde{U}_i^{(t)}$ accumulates directions used during the original training of $\mathcal{T}_i$ as well as all subsequent backward refinements. This mechanism ensures that each refinement step continues to explore only previously unused directions, while remaining non-interfering over time, as shown by the following result, whose proof can be found in Appendix §A.

**Proposition 4.1** (Cumulative non-reuse and non-interference). *Fix a prior task $\mathcal{T}_i$ with prompt $u_i \in \mathbb{R}^{\ell \times d}$, and let $\tilde{U}_i^{(t)} \in \mathbb{R}^{(\ell d) \times r_i^{(t)}}$ denote an orthonormal basis of the protected subspace at time $t$, initialized as $\tilde{U}_i^{(i)} = U_i$. For any refinement time $t > i$, define the safe update $\Delta u_i^{(t)} = \big(I - \tilde{U}_i^{(t-1)}\tilde{U}_i^{(t-1)\top}\big)\nabla_{u_i}\mathcal{L}_t(u_i)$, and update the protected subspace as $\tilde{U}_i^{(t)} \leftarrow \mathrm{orth}([\tilde{U}_i^{(t-1)}, \Delta u_i^{(t)}/\|\Delta u_i^{(t)}\|_2])$ when $\Delta u_i^{(t)} \neq 0$ (otherwise set $\tilde{U}_i^{(t)} = \tilde{U}_i^{(t-1)}$). For all $t \geq i$:*

1. *(**Cumulative span**) $\mathrm{span}(\tilde{U}_i^{(t)})$ contains $\mathrm{span}(U_i)$ and all previously appended refinement directions.*

2. *(**Non-interference**) $\tilde{U}_i^{(t-1)\top}\Delta u_i^{(t)} = 0$, i.e., each refinement update is orthogonal to all directions protected from original training and earlier refinements.*

### 4.3. Prompt-Based Continual Learning with Selective Backward Refinement

We now integrate the aforementioned components and propose **S**elective b**A**ckward refinement for positive **B**ackward knowledge transf**ER** (SABER) for prompt-based continual learning. The overall procedure is summarized in Algorithm 1. Concretely, for each task $T_t$, we introduce a task-specific prompt $u_t$ and optimize it via standard gradient descent while keeping the backbone frozen. For the first task, SABER reduces to standard prompt tuning. For subsequent tasks $t \geq 2$, we identify a subset of prior tasks that are sufficiently correlated with the current task according to one of the following **two criteria** (C.1 or C.2):

$$S_t = \left\{\ i \in \{1,\ldots,t-1\} \ \middle|\ \begin{array}{l} \text{C.1: } s_i \geq \tau_s \ \wedge \ c_i > 0 \\ \text{C.2: } \mathrm{WSS}_{t,i} \geq \tau_{\mathrm{WSS}} \end{array}\ \right\}$$

where $s_i$ denotes the projection score, $c_i$ the gradient compatibility, and $\mathrm{WSS}_{t,i}$ the Wasserstein similarity score. Importantly, SABER uses *fixed global thresholds* $\tau_s = 0.1$ and $\tau_{\mathrm{WSS}} = 0.2$ throughout all experiments; we find these values to be robust across datasets and model settings, requiring no per-benchmark tuning (see Appendix for details). Only prompts indexed by $S_t$ are refined, while all other prior prompts remain frozen during training of $T_t$. In practice, we first optimize the current-task prompt $u_t$, and perform backward refinement only after $u_t$ has sufficiently stabilized.

For each selected $i \in S_t$, the backward update is constrained to avoid interference with protected directions by enforcing $\tilde{U}_i^{(t-1)\top}\Delta u_i^{(t)} = 0$. The learning objective for task $T_t$ can be summarized as

$$\min_{\{u_t\} \cup \{u_i\}_{i \in S_t}} \mathcal{L}_t(u_t)$$
$$\text{s.t.}\quad \tilde{U}_i^{(t-1)\top}\Delta u_i^{(t)} = 0, \qquad \forall i \in S_t.$$

Overall, SABER enables controlled backward knowledge transfer to selected prior prompts without replay, while mitigating negative transfer by explicitly preventing updates

---

**Algorithm 1** Selective Backward Refinement (SABER)

---

**Require:** Tasks $\{(\mathcal{T}_t, \mathcal{D}_t)\}_{t=1}^T$, frozen LM $f(\cdot; \theta)$, rank $r$, thresholds $\tau_s$ or $\tau_{\text{WSS}}$
**output** Prompts $\{u_t\}_{t=1}^T$ and protected bases $\{\tilde{U}_t\}_{t=1}^T$
1: $\mathcal{U}_0 \leftarrow \emptyset$
2: **for** $t = 1$ **to** $T$ **do**
3:     **if** $t > 1$ **then**
4:         Select correlated indices $I_t \subseteq \{1, \ldots, t-1\}$ using projection / loss-distribution criterion {Eq. 1, 2, 3}
5:     **else**
6:         $I_t \leftarrow \emptyset$
7:     **end if**
8:     Train current prompt $u_t$ on $\mathcal{D}_t$ with $\theta$ frozen
9:     $U_t \leftarrow \text{GradSubspaceSVD}(\mathcal{D}_t, u_t, r)$ {Appendix B}
10:     $\tilde{U}_t \leftarrow U_t$; $\mathcal{U}_t \leftarrow \mathcal{U}_{t-1} \cup \{u_t\}$
11:     **for** each $i \in I_t$ **do**
12:         **for** $k = 1$ **to** $K$ **do** {few safe refinement steps}
13:             $\Delta u_i \leftarrow (I - \tilde{U}_i \tilde{U}_i^\top) \nabla_{u_i} L_t(u_i)$ {Eq. 4}
14:             $u_i \leftarrow u_i - \eta \, \Delta u_i$
15:         **end for**
16:         $\tilde{U}_i \leftarrow \text{orth}(\tilde{U}_i, \Delta u_i)$ {Eq. 6}
17:     **end for**
18: **end for**

---

along task-critical directions. The following result shows that our backward refinement is both safe and effective.

**Proposition 4.2** (Monotone improvement under $K$ safe refinement steps)**.** *Fix a prior prompt $u_i$ at time $t$, and let $\tilde{U} := \tilde{U}_i^{(t-1)}$ be a matrix with orthonormal columns. Define $f(u) := \mathcal{L}_t(u)$ and the safe (projected) gradient $\Delta(u) := (I - \tilde{U}\tilde{U}^\top)\nabla f(u)$, which removes all components along the protected subspace $\text{span}(\tilde{U})$. Starting from $u^{(0)} = u_i$, we perform $K$ refinement steps $u^{(k+1)} = u^{(k)} - \eta \, \Delta(u^{(k)})$ for $k = 0, \ldots, K - 1$. Assume that $f$ is $L$-smooth in $u$ and the stepsize satisfies $\eta \leq 1/L$. Then for any $K \geq 1$,*

$$f(u^{(0)}) - f(u^{(K)}) \geq \frac{\eta}{2} \sum_{k=0}^{K-1} \left\| \Delta(u^{(k)}) \right\|_2^2,$$

*and therefore $f(u^{(K)}) \leq f(u^{(0)})$, with strict inequality whenever $\Delta(u^{(k)}) \neq 0$ for some $k < K$. In other words, each safe refinement step is guaranteed to be non-increasing on the current-task loss, while never updating directions inside the protected subspace.*

*Moreover, $\Delta(u^{(0)}) = 0$ if and only if $\nabla f(u^{(0)}) \in \text{span}(\tilde{U})$. In this case, $\langle \nabla f(u^{(0)}), d \rangle = 0$ for every direction $d$ satisfying $\tilde{U}^\top d = 0$, i.e., there is no first-order descent direction that preserves the protected subspace.*

## 5. Experiment

### 5.1. Experiment setups

#### Dataset and Metrics.

We evaluate our method on two standard task-incremental continual learning benchmarks, SUPERNI (Wang et al.,

2022a) and LONG SEQUENCE (Razdaibiedina et al., 2023), each comprising 15 sequential tasks and evaluated under two task orders following prior work (Razdaibiedina et al., 2023; Zhao et al., 2024). Let $\text{Acc}_{\mathcal{T}_t, \mathcal{T}_k}$ denote the test performance on task $\mathcal{T}_k$ after training on task $\mathcal{T}_t$, measured by accuracy for classification tasks and ROUGE-L (Chin-Yew, 2004) otherwise. We report three standard continual learning metrics: *Average Performance* (AP) (Chaudhry et al., 2018a), $\text{AP} = \frac{1}{T} \sum_{t=1}^T \text{Acc}_{\mathcal{T}_T, \mathcal{T}_t}$; *Forward Transfer* (FWT) (Lopez-Paz & Ranzato, 2017); and *Backward Transfer* (BWT) (Ke & Liu, 2022). Further dataset details are provided in Appendix §D.

**Comparison Baselines and Training Details.** We compare our method against seven continual learning baselines, including Replay (de Masson D'Autume et al., 2019), L2P (Wang et al., 2022c), LFPT5 (Qin & Joty, 2022), Prog-Prompt (Razdaibiedina et al., 2023), CODA (Smith et al., 2023), SHLPT (Wu et al., 2024), and SAPT (Zhao et al., 2024). All methods are evaluated under a task-incremental setting using the same pretrained backbone models for fairness. Our method is a model-agnostic continual learning framework compatible with transformer-based pretrained models; in our experiments, we use diverse models such as T5, Qwen, and LLaMA. We report two variants of our method: SABER-P, which uses projection-based task correlation, and SABER-L, which uses loss-distribution-based task correlation. Additional implementation details and hyperparameter settings are provided in Appendix §E.

**Frameworks.** We evaluate our method under two major prompt-based continual learning frameworks that differ in how prior task knowledge is used during training of the current task. In the *Frozen Prompt Pool (FPP) Framework*, each task $T_t$ has a task-specific prompt $u_t$, and all previously learned prompts $\mathcal{U}_{t-1}$ are used during training to support forward transfer; at task $t$, the current prompt $u_t$ and a selected subset of prior prompts are trainable for backward refinement, while the remaining prompts are kept frozen. In contrast, the *Shared Prompt Augmentation (SPA) Framework* supports forward transfer through a shared prompt $P$ with larger capacity that is updated across tasks; at task $t$, training uses only the shared prompt $P$, the current prompt $u_t$, and selected prior prompts for backward refinement, all updated with distinct learning rates.

### 5.2. Main Results

Tables 5 and 6 report results on the Long Sequence and SuperNI benchmarks. Across all settings, SABER consistently outperforms strong prompt-based continual learning baselines, improving average performance and, importantly, achieving positive backward transfer. In contrast, all competing methods exhibit negative BWT, indicating performance degradation on earlier tasks. SABER is evaluated under

*Table 5.* Results on Long Sequence and SuperNI for `T5-Large`. Higher is better for both metrics. Positive BWT indicates improvement on earlier tasks after learning subsequent tasks. **Bold** indicates the best result and underline indicates the second best.

| Method | Long Sequence | | | | SuperNI | | | |
|---|---|---|---|---|---|---|---|---|
| | Order1 | | Order2 | | Order1 | | Order2 | |
| | AP↑ | BWT↑ | AP↑ | BWT↑ | AP↑ | BWT↑ | AP↑ | BWT↑ |
| Replay | 56.20±0.74 | -12.18±0.63 | 53.50±0.81 | -17.18±0.72 | 33.65±0.68 | -14.20±0.59 | 35.70±0.77 | -16.42±0.66 |
| L2P | 60.43±0.61 | -1.43±0.34 | 59.87±0.58 | -0.83±0.29 | 14.32±0.52 | -1.54±0.31 | 13.84±0.49 | -1.65±0.35 |
| LFPT5 | 69.35±0.66 | -0.76±0.27 | 67.64±0.71 | -0.53±0.22 | 36.43±0.73 | -0.32±0.19 | 35.98±0.69 | -0.43±0.21 |
| ProgPrompt | 75.43±0.55 | -0.12±0.31 | 74.30±0.62 | -0.37±0.26 | 38.84±0.64 | -0.27±0.28 | 39.32±0.59 | -0.21±0.33 |
| CODA Prompt | 76.21±0.60 | -0.54±0.25 | 75.95±0.58 | -0.65±0.27 | 42.34±0.57 | -0.98±0.30 | 42.76±0.61 | -0.87±0.29 |
| SHLPT | 77.40±0.53 | -0.29±0.28 | 76.87±0.56 | -0.34±0.26 | 43.87±0.59 | -0.11±0.27 | 44.32±0.62 | -0.20±0.24 |
| SAPT | 78.14±0.48 | -0.45±0.21 | 77.54±0.51 | -0.65±0.24 | 43.76±0.55 | -0.54±0.22 | 41.21±0.58 | -0.98±0.29 |
| **FPP + SABER-P** | **80.46**±0.44 | 1.76±0.18 | **80.21**±0.47 | 2.15±0.21 | 45.50±0.49 | 1.86±0.20 | **47.43**±0.52 | 2.32±0.23 |
| **FPP + SABER-L** | 80.12±0.54 | **2.13**±1.18 | 79.94±0.49 | **3.12**±0.31 | **46.08**±0.49 | **1.90**±0.50 | 45.75±1.52 | **2.88**±0.12 |
| **SPA + SABER-P** | **80.84**±0.46 | 0.93±0.19 | 78.68±0.50 | 1.29±0.22 | 45.21±0.48 | 1.22±0.21 | 45.13±0.51 | 1.45±0.24 |
| **SPA + SABER-L** | 80.67±1.47 | 0.87±0.36 | 78.17±1.34 | 1.65±0.92 | 44.28±0.23 | 1.12±0.45 | 46.12±0.48 | 1.14±0.33 |

*Table 6.* Results on Long Sequence and SuperNI for `LLaMA-2-7B`. Higher is better for both metrics. Positive BWT indicates improvement on earlier tasks after learning subsequent tasks. **Bold** indicates the best result and underline indicates the second best.

| Method | Long Sequence | | | | SuperNI | | | |
|---|---|---|---|---|---|---|---|---|
| | Order1 | | Order2 | | Order1 | | Order2 | |
| | AP↑ | BWT↑ | AP↑ | BWT↑ | AP↑ | BWT↑ | AP↑ | BWT↑ |
| Replay | 60.32±0.71 | -19.54±0.83 | 59.98±0.68 | -21.09±0.77 | 37.48±0.62 | -21.47±0.69 | 40.23±0.74 | -23.32±0.81 |
| LFPT | 72.65±0.54 | -0.94±0.29 | 71.87±0.61 | -0.56±0.24 | 38.88±0.57 | -0.98±0.31 | 38.95±0.63 | -0.67±0.26 |
| ProgPrompt | 78.98±0.49 | -0.18±0.33 | 77.33±0.52 | -0.12±0.21 | 40.65±0.55 | -0.26±0.28 | 43.98±0.58 | -0.17±0.30 |
| SHLPT | 79.40±0.47 | -0.27±0.27 | 80.65±0.50 | -0.15±0.23 | 44.97±0.53 | -0.45±0.25 | 46.97±0.56 | -0.33±0.22 |
| SAPT | 78.43±0.51 | -0.86±0.35 | 78.98±0.48 | -0.85±0.29 | 46.98±0.59 | -0.75±0.24 | 47.42±0.61 | -0.43±0.21 |
| **FPP + SABER-P** | **82.87**±0.44 | 1.56±0.18 | 83.98±0.46 | 1.95±0.21 | 48.65±0.49 | 2.13±0.20 | **49.26**±0.52 | **1.98**±0.22 |
| **FPP + SABER-L** | 82.23±0.53 | 1.12±0.24 | **84.86**±0.51 | **2.11**±0.26 | 47.12±0.58 | 1.90±0.31 | 47.10±0.60 | 0.12±0.19 |
| **SPA + SABER-P** | 81.47±0.48 | 1.39±0.22 | 79.84±0.50 | 1.94±0.27 | **49.48**±0.46 | 2.18±0.24 | 47.65±0.55 | 0.97±0.28 |
| **SPA + SABER-L** | 82.65±0.45 | **1.87**±0.21 | 79.34±0.52 | 1.33±0.25 | 49.21±0.50 | **2.75**±0.29 | 46.87±0.57 | 1.13±0.26 |

both the Frozen Prompt Pool (FPP) and Shared Prompt Augmentation (SPA) frameworks, achieving the best BWT in both cases. FPP generally yields larger backward transfer gains, while SPA maintains consistently positive BWT with competitive AP. Both correlation variants, **SABER-P** and **SABER-L**, show consistent improvements, suggesting that backward transfer can be enabled using either update-level or response-level task correlation signals. Figure 4 and Figure 5 provides a task-level view, illustrating accuracy improvements induced by backward refinement.

*Table 7.* Performance on `Qwen-3-4B` on the Long Sequence benchmark. We report only the strongest ProgPrompt and SHLPT baselines and our projection-based SABER version for brevity.

| Method | Long Sequence | | | |
|---|---|---|---|---|
| | Order 1 | | Order 2 | |
| | AP | BWT | AP | BWT |
| ProgPrompt | 79.70±0.42 | -0.18±0.19 | 78.32±0.47 | -0.28±0.17 |
| SHLPT | 80.95±0.38 | -0.27±0.21 | 80.54±0.41 | -0.23±0.18 |
| **FPP + SABER-P** | **83.98**±0.36 | **2.14**±0.22 | **83.32**±0.44 | 1.96±0.25 |
| **SPA + SABER-P** | 81.97±0.40 | 1.87±0.24 | 82.68±0.39 | **2.12**±0.27 |

**Generalization and Scalability.** Table 6 shows that these trends persist when scaling from `T5-Large` (0.77B) to `LLaMA-2-7B`, where SABER again achieves the strongest AP and is the only method with consistently positive BWT. Table 7 further extends these results to `Qwen-3-4B`, where SABER yields larger backward transfer gains while maintaining strong overall performance.

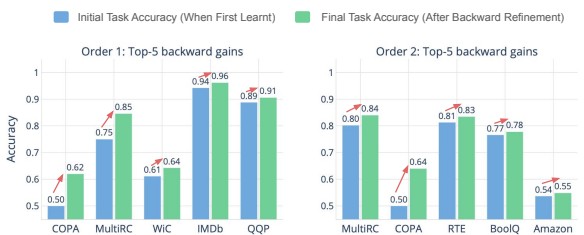

*Figure 4.* Task-level backward refinement under FPP on the Long Sequence benchmark. Bars show initial accuracy when each task is first learned and final accuracy after completing the task sequence; arrows indicate accuracy gains from backward refinement.

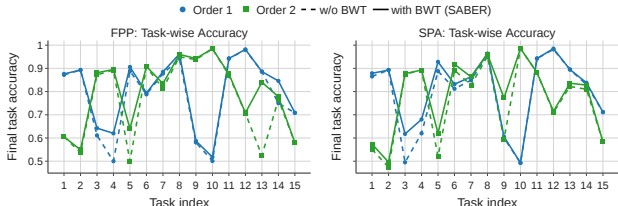

*Figure 5.* Task-wise final accuracy on the Long Sequence benchmark with (solid line) and without (dashed line) backward refinement. Results are shown for FPP + SABER-P (left) and SPA + SABER-L (right).

*Table 8.* Runtime overhead of SABER over the full task sequence. We report total GPU time and additional overhead relative to the corresponding standard prompt-based baseline.

| Backbone | Method | Total GPU Time (h:m) | Overhead |
|---|---|---|---|
| T5-Large | SABER-P | 24:09 | +06:12 |
| T5-Large | SABER-L | 26:00 | +08:03 |
| LLaMA-2-7B | SABER-P | 105:04 | +32:00 |
| LLaMA-2-7B | SABER-L | 100:08 | +27:04 |

### 5.3. Runtime and Computational Overhead

To assess the practical cost of SABER, we report total wall-clock training time over the full 15-task sequence on a single NVIDIA A100 80GB GPU. As shown in Table 8, SABER adds moderate overhead relative to the corresponding prompt-based baseline: about $34\%$ for T5-Large and $44\%$ for LLaMA-2-7B, while being the only method to achieve consistently positive BWT. This overhead comes from task-correlation estimation and safe backward refinement of selected prior prompts. The refinement cost scales as $\mathcal{O}(|\mathcal{S}_t|K\ell dr)$, where $|\mathcal{S}_t|$ is the number of selected prompts, $K$ the number of refinement steps, $\ell d$ the prompt dimension, and $r$ the protected-subspace rank.

The two variants offer different efficiency trade-offs. SABER-P uses gradient subspaces, providing finer geometric control but incurring memory growth with the number of tasks. In contrast, SABER-L uses scalar loss statistics, reducing memory overhead by up to $2,340$ KB at 15 tasks (Figure 6, Appendix §Table 15) while adding only modest extra compute. Thus, SABER-L is more memory-efficient for large-scale deployment, whereas SABER-P is preferable when stronger geometric control is desired. A summary of all quantities stored per prior task is provided in Appendix §Table 16.

### 5.4. Ablation Study

Table 9 presents an ablation study on Long Sequence and SuperNI. Removing *task correlation* causes backward trans-

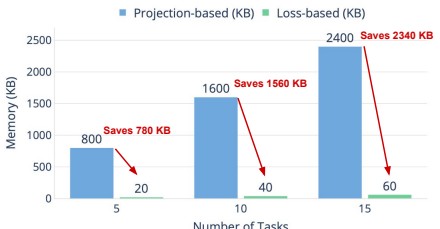

*Figure 6.* Memory overhead comparison between projection-based and loss-distribution-based task correlation.

fer to collapse, yielding negative BWT, which shows that indiscriminate backward updates are often harmful. Removing *constrained backward refinement* also degrades performance, particularly BWT, indicating that correlated tasks alone are insufficient without constrained updates. Removing both components leads to the worst performance, with the lowest AP and most negative BWT. Overall, task correlation and constrained backward refinement are complementary and jointly necessary for stable backward transfer.

*Table 9.* Ablation study of SABER.

| Model | Variant | Long Sequence | | SuperNI | |
|---|---|---|---|---|---|
| | | AP | BWT | AP | BWT |
| | **(0) SABER** | **80.33** | **1.95** | **42.48** | **2.09** |
| T5-Large | (1) w/o task correlation | 77.59 | -1.78 | 40.93 | -0.84 |
| | (2) w/o constrained refinement | 78.84 | -0.75 | 41.33 | -0.43 |
| | (3) w/o all | 77.02 | -1.98 | 39.28 | -1.35 |
| | **(0) SABER** | **83.42** | **1.75** | **48.95** | **2.05** |
| LLaMA | (1) w/o task correlation | 81.26 | -1.38 | 45.84 | -0.89 |
| | (2) w/o constrained refinement | 80.54 | -0.87 | 46.98 | -0.74 |
| | (3) w/o all | 79.84 | -1.82 | 45.12 | -1.16 |

## 6. Conclusion

We introduced SABER, a replay-free framework that enables selective and safe backward knowledge transfer in prompt-based continual learning. By explicitly identifying when backward refinement is beneficial and constraining updates to non-interfering prompt directions, SABER achieves consistent positive backward transfer while preserving strong overall performance. Our theoretical analysis guarantees interference-free refinements under mild conditions, and extensive experiments across models and benchmarks validate the effectiveness and efficiency of the approach. Future work includes extending SABER to more complex continual learning settings such as longer task sequences and mixed task similarities, exploring richer task-correlation measures beyond gradients and loss distributions, and integrating selective backward refinement with other parameter-efficient adaptation strategies.

## Acknowledgements

The work of A. Tiwari and K. Ji was partially supported by NSF grants CCF-2311274, ECCS-2326592 and CAREER-2442418.

## Impact Statement

This paper presents work whose primary goal is to advance the field of machine learning by improving parameter-efficient continual learning methods. While the proposed approach may enable more efficient and adaptive model updates in practice, it does not introduce new application domains or capabilities beyond those of existing pretrained language models. Any broader societal impacts are therefore expected to be similar to those of current large language models, and we do not identify specific ethical concerns or societal consequences that require additional discussion beyond standard responsible use considerations.

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

# Appendix

## A. Proposition

### Proof of Proposition 4.1

*Proof.* **Part (1).** We prove the cumulative span claim by induction on $t$. For $t = i$, $\tilde{U}_i^{(i)} = U_i$, hence $\mathrm{span}(\tilde{U}_i^{(i)}) = \mathrm{span}(U_i)$. Assume the claim holds at time $t - 1$. If $\Delta u_i^{(t)} = 0$, then $\tilde{U}_i^{(t)} = \tilde{U}_i^{(t-1)}$ and the claim follows immediately. Otherwise, $\Delta u_i^{(t)} \neq 0$ and $\tilde{U}_i^{(t)} = \mathrm{orth}([\tilde{U}_i^{(t-1)}, \Delta u_i^{(t)}/\|\Delta u_i^{(t)}\|_2])$. By definition of $\mathrm{orth}(\cdot)$, it returns an orthonormal basis spanning the same column space, so

$$\mathrm{span}(\tilde{U}_i^{(t)}) = \mathrm{span}(\tilde{U}_i^{(t-1)}) + \mathrm{span}\left(\Delta u_i^{(t)}\right).$$

Thus $\mathrm{span}(\tilde{U}_i^{(t)})$ contains $\mathrm{span}(\tilde{U}_i^{(t-1)})$ and the newly appended refinement direction; by the induction hypothesis, it contains $\mathrm{span}(U_i)$ and all previously appended directions.

**Part (2).** Let $g_i^{(t)} = \nabla_{u_i} \mathcal{L}_t(u_i)$. Using the update definition and orthonormality of $\tilde{U}_i^{(t-1)}$,

$$\tilde{U}_i^{(t-1)\top} \Delta u_i^{(t)} = \tilde{U}_i^{(t-1)\top}\left(I - \tilde{U}_i^{(t-1)}\tilde{U}_i^{(t-1)\top}\right)g_i^{(t)}$$
$$= \left(\tilde{U}_i^{(t-1)\top} - \tilde{U}_i^{(t-1)\top}\right)g_i^{(t)} = 0.$$

Hence $\Delta u_i^{(t)} \perp \mathrm{span}(\tilde{U}_i^{(t-1)})$. By Part (1), this span contains all directions protected from original training and earlier refinements, implying that each backward update is non-interfering. $\square$

### Proof of Proposition 4.3

*Proof.* Let $P := \tilde{U}\tilde{U}^\top$. Since $\tilde{U}^\top \tilde{U} = I$, $P$ is an orthogonal projector, so $(I - P)$ is symmetric and idempotent. For each iterate, write $g^{(k)} := \nabla f(u^{(k)})$ and $\Delta^{(k)} := (I - P)g^{(k)}$. Then

$$\langle g^{(k)}, \Delta^{(k)} \rangle = \langle g^{(k)}, (I - P)g^{(k)} \rangle = \langle (I - P)g^{(k)}, (I - P)g^{(k)} \rangle = \|\Delta^{(k)}\|_2^2.$$

By $L$-smoothness (descent lemma), with $d = -\eta\Delta^{(k)}$,

$$f(u^{(k+1)}) \leq f(u^{(k)}) + \langle g^{(k)}, d \rangle + \frac{L}{2}\|d\|_2^2 = f(u^{(k)}) - \eta\|\Delta^{(k)}\|_2^2 + \frac{L\eta^2}{2}\|\Delta^{(k)}\|_2^2.$$

Thus

$$f(u^{(k)}) - f(u^{(k+1)}) \geq \eta\left(1 - \frac{L\eta}{2}\right)\|\Delta^{(k)}\|_2^2 \geq \frac{\eta}{2}\|\Delta^{(k)}\|_2^2,$$

where the last inequality uses $\eta \leq 1/L$. Summing over $k = 0, \ldots, K - 1$ yields

$$f(u^{(0)}) - f(u^{(K)}) \geq \frac{\eta}{2}\sum_{k=0}^{K-1}\|\Delta^{(k)}\|_2^2,$$

which implies $f(u^{(K)}) \leq f(u^{(0)})$, with strict inequality if some $\Delta^{(k)} \neq 0$.

Finally, $\Delta(u^{(0)}) = (I - P)\nabla f(u^{(0)}) = 0$ iff $\nabla f(u^{(0)}) \in \mathrm{range}(P) = \mathrm{span}(\tilde{U})$. In this case, for any $d$ with $\tilde{U}^\top d = 0$ (equivalently $Pd = 0$), we have $\langle \nabla f(u^{(0)}), d \rangle = \langle P\nabla f(u^{(0)}), d \rangle = \langle \nabla f(u^{(0)}), Pd \rangle = 0$. $\square$

## B. Construction of Gradient Subspace via SVD

We detail the construction of the task-specific gradient subspace basis $U_i \in \mathbb{R}^{(\ell d) \times r}$ used in Section 3.

**Gradient Matrix.** For a task $\mathcal{T}_i$ with prompt $u_i \in \mathbb{R}^{\ell \times d}$, let $\nabla_{u_i} \mathcal{L}_i(B; u_i) \in \mathbb{R}^{\ell \times d}$ denote the gradient of the task loss with respect to the prompt, evaluated on a minibatch $B \sim \mathcal{D}_i$. We vectorize this gradient as

$$g_i(B) := \mathrm{vec}(\nabla_{u_i} \mathcal{L}_i(B; u_i)) \in \mathbb{R}^{\ell d}.$$

Collecting gradients over $N_i$ minibatches $\{B_j\}_{j=1}^{N_i}$, we form the task gradient matrix

$$G_i := \begin{bmatrix} g_i(B_1) & g_i(B_2) & \cdots & g_i(B_{N_i}) \end{bmatrix} \in \mathbb{R}^{(\ell d) \times N_i}.$$

**Singular Value Decomposition.** We compute the singular value decomposition (SVD) of $G_i$:

$$G_i = U_i^{\text{full}} \Sigma_i V_i^\top,$$

where $U_i^{\text{full}} \in \mathbb{R}^{(\ell d) \times (\ell d)}$ contains the left singular vectors, $\Sigma_i$ is a diagonal matrix of singular values $\sigma_1 \geq \sigma_2 \geq \cdots \geq 0$, and $V_i \in \mathbb{R}^{N_i \times N_i}$ contains the right singular vectors.

**Low-Rank Gradient Subspace.** We retain the top-$r$ left singular vectors corresponding to the largest singular values, where $r \ll \ell d$, and define

$$U_i := \begin{bmatrix} u_1 & u_2 & \cdots & u_r \end{bmatrix} \in \mathbb{R}^{(\ell d) \times r}.$$

By construction, the columns of $U_i$ are orthonormal and span the dominant gradient subspace of task $\mathcal{T}_i$, capturing the directions along which updates to the prompt were most significant during training.

**Interpretation.** The subspace $\text{span}(U_i)$ provides a low-dimensional approximation of the task's gradient space. Projecting gradients from a new task onto this subspace yields a geometry-aware measure of update alignment across tasks, which we use both for task correlation estimation and for defining protected directions during backward refinement.

## C. Threshold Sensitivity Analysis

SABER uses two global thresholds: the projection threshold $\tau_s$ for update-level alignment (SABER-P) and the Wasserstein similarity threshold $\tau_{\text{WSS}}$ for response-level alignment (SABER-L). In all experiments, we fix $\tau_s = 0.1$ and $\tau_{\text{WSS}} = 0.2$ and apply the same values across all datasets and task orders, requiring no per-benchmark tuning.

Table 10 reports AP and BWT on the Long Sequence benchmark (T5-Large, Order 1) across a range of threshold values. Very small thresholds select too many prior tasks for refinement, increasing negative transfer and hurting both AP and BWT. Conversely, very large thresholds are overly conservative, selecting too few prior tasks and reducing backward-transfer gains. The best results consistently occur in a moderate range around the default values, with SABER-P remaining stable within $\tau_s \in [0.05, 0.20]$ and SABER-L within $\tau_{\text{WSS}} \in [0.15, 0.30]$, yielding similar task selections and comparable performance across this range. These results indicate that SABER is reasonably robust and does not require fine-grained threshold tuning.

*Table 10.* Sensitivity of AP and BWT to threshold values on Long Sequence (T5-Large, Order 1). Bold indicates the chosen default value.

*(a)* SABER-P: varying $\tau_s$

| $\tau_s$ | 0.01 | 0.05 | **0.10** | 0.20 | 0.40 | 0.60 |
|---|---|---|---|---|---|---|
| AP | 78.1 | 79.8 | **80.3** | 80.1 | 78.9 | 77.2 |
| BWT | -0.8 | 1.2 | **2.1** | 1.9 | 1.2 | 0.4 |

*(b)* SABER-L: varying $\tau_{\text{WSS}}$

| $\tau_{\text{WSS}}$ | 0.05 | 0.10 | **0.20** | 0.35 | 0.50 | 0.70 |
|---|---|---|---|---|---|---|
| AP | 77.5 | 79.2 | **80.3** | 80.0 | 79.1 | 77.8 |
| BWT | -1.1 | 1.0 | **1.9** | 1.8 | 1.1 | 0.3 |

## D. Additional Dataset and Metric Details

**Dataset.** Following prior work (Li et al., 2026), we adopt **SuperNI** and **Long Sequence** as benchmark suites for evaluating continual learning methods in large language models (LLMs). Detailed task descriptions and evaluation metrics are summarized in Table 11, and two distinct task orders are considered for each benchmark, as shown in Table 12. The **SuperNI** benchmark (Wang et al., 2022a) comprises a diverse collection of NLP tasks paired with expert-written instructions, enabling rigorous and realistic evaluation in continual learning settings. It consists of 15 sequential tasks. For each task,

*Table 11.* Details of the SuperNI and Long Sequence benchmarks.

| SuperNI Benchmark | | | Long Sequence Benchmark | | |
|---|---|---|---|---|---|
| **Dataset name** | **Task** | **Metric** | **Dataset name** | **Task** | **Metric** |
| 1. task639 | dialogue generation | Rouge-L | 1. Yelp | sentiment analysis | accuracy |
| 2. task1590 | dialogue generation | Rouge-L | 2. Amazon | sentiment analysis | accuracy |
| 3. task1729 | dialogue generation | Rouge-L | 3. DBpedia | topic classification | accuracy |
| 4. task181 | information extraction | Rouge-L | 4. Yahoo | topic classification | accuracy |
| 5. task748 | information extraction | Rouge-L | 5. AG News | topic classification | accuracy |
| 6. task1510 | information extraction | Rouge-L | 6. MNLI | natural language inference | accuracy |
| 7. task002 | question answering | Rouge-L | 7. QQP | paraphrase detection | accuracy |
| 8. task073 | question answering | Rouge-L | 8. RTE | natural language inference | accuracy |
| 9. task591 | question answering | Rouge-L | 9. SST-2 | sentiment analysis | accuracy |
| 10. task511 | summarization | Rouge-L | 10. WiC | word sense disambiguation | accuracy |
| 11. task1290 | summarization | Rouge-L | 11. CB | natural language inference | accuracy |
| 12. task1572 | summarization | Rouge-L | 12. COPA | question answering | accuracy |
| 13. task363 | sentiment analysis | accuracy | 13. BoolQA | boolean question answering | accuracy |
| 14. task875 | sentiment analysis | accuracy | 14. MultiRC | question answering | accuracy |
| 15. task1687 | sentiment analysis | accuracy | 15. IMDB | sentiment analysis | accuracy |

*Table 12.* Different task order of Long Sequence and SuperNI benchmark.

| Order | SuperNI Benchmark | Long Sequence Benchmark |
|---|---|---|
| 1 | task1572 → task363 → task1290 → task181 → task002 → task1510 → task639 → task1729 → task073 → task1590 → task748 → task511 → task591 → task1687 → task875 | MNLI → CB → WiC → COPA → QQP → BoolQA → RTE → IMDB → Yelp → Amazon → SST-2 → DBpedia → AG News → MultiRC → Yahoo |
| 2 | task748 → task073 → task1590 → task639 → task1572 → task1687 → task591 → task363 → task1510 → task1729 → task181 → task511 → task002 → task1290 → task875 | Yelp → Amazon → MNLI → CB → COPA → QQP → RTE → IMDB → SST-2 → DBpedia → AG News → Yahoo → MultiRC → BoolQA → WiC |

1,000 instances are randomly sampled for training, while 100 instances are used for validation and testing. The **Long Sequence** benchmark (Razdaibiedina et al., 2023) also contains 15 classification tasks, specifically designed to evaluate continual learning with LLMs under long-context scenarios. For each task, 1,000 samples are used for training, and 500 samples per class are allocated for both validation and testing. This benchmark highlights challenges associated with long-context dependencies and task diversity in sequential learning.

**Metrics.** Let $\text{Acc}_{\mathcal{T}_t, \mathcal{T}_k}$ denote the test performance on task $\mathcal{T}_k$ after training on task $\mathcal{T}_t$, measured by accuracy for classification tasks and ROUGE-L (Chin-Yew, 2004) for other tasks. We report three standard continual learning metrics: *Average Performance* (AP) (Chaudhry et al., 2018a), defined as the mean performance over all tasks after training on the final task, $\text{AP} = \frac{1}{T} \sum_{t=1}^{T} \text{Acc}_{\mathcal{T}_T, \mathcal{T}_t}$; *Forward Transfer* (FWT) (Lopez-Paz & Ranzato, 2017), which quantifies the effect of prior tasks on learning new tasks; and *Backward Transfer* (BWT) (Ke & Liu, 2022), which measures the impact of learning new tasks on previously learned tasks.

# E. Additional Training Details

**Training Details.** All experiments are conducted in a continual learning setting with a frozen backbone model. We consider both autoregressive and sequence-to-sequence architectures, with training configurations summarized in Table 13. For autoregressive models, we use Qwen3-4B-Instruct and LLaMA-2-7B as backbones, while T5-large is adopted for sequence-to-sequence models. Each benchmark consists of 15 sequential tasks learned one at a time using task-specific soft prompts. Across both architectures, the prompt prefix length is set to 10 and the maximum input sequence length is capped at 256 tokens. We use a learning rate of 0.03 for autoregressive backbones and 0.3 for sequence-to-sequence backbones, with batch sizes of 16 and 8, respectively. Each task is trained for 5 epochs in the autoregressive setting and 10 epochs in the sequence-to-sequence setting, with early stopping based on validation performance enabled for all experiments. For each

*Table 13.* Training configurations used in our experiments for autoregressive and seq-to-seq backbones.

| Argument | Autoregressive | Seq-to-seq |
|---|---|---|
| Architecture | Autoregressive | Seq-to-seq |
| Base model | Qwen3-4B-Instruct; LLaMA-2-7B | T5-Large |
| Number of tasks | 15 | 15 |
| Prefix length | 10 | 10 |
| Max sequence length | 256 | 256 |
| Learning rate | 0.03 | 0.3 |
| Batch size | 16 | 8 |
| Samples per class ($k$) | 1000 | 1000 |
| Training epochs | 5 | 10 |
| Frozen backbone | Yes | Yes |
| Early stopping | Enabled | Enabled |
| Protected-subspace rank ($r$) | 3 | 3 |
| Backward refinement LR | 0.001 | 0.001 |
| Backward refinement schedule | Final 2 epochs | Final 2 epochs |

task, 1,000 training samples are used, with fixed validation and test splits to ensure consistent evaluation across methods. We use protected-subspace rank $r = 3$ across all experiments. We found larger ranks, such as $r = 5$, to give comparable or slightly worse performance while increasing memory overhead. Backward refinement is performed only during the final two epochs of current-task training, after the current-task prompt has stabilized, using a smaller learning rate of 0.001. For the Replay baseline, we follow prior work and use a replay buffer of 100 samples per task. All reported results are averaged over three independent runs.

**Prompt tuning and initialization.** For all architectures, we freeze the backbone parameters and optimize only task-specific soft prompts. For instruction-tuned autoregressive models, prompts are initialized using a short natural-language instruction that enumerates task label names (e.g., "Classify into: . . . "), and the corresponding token embeddings are used to seed the 10-token soft prompt. For sequence-to-sequence models, prompts are initialized directly in the embedding space. We also experimented with random prompt initialization and observed a slight but not statistically significant degradation in performance compared to instruction-based initialization.

**Runtime and Hardware.** All experiments are conducted on a single NVIDIA A100 GPU with 80 GB memory per run. We implement all methods in PyTorch and report GPU runtime as total wall-clock training time over the full task sequence, along with the additional overhead relative to the corresponding prompt-based baseline without backward refinement under the same framework (FPP or SPA). This overhead isolates the extra computation introduced by task-correlation estimation and backward refinement. Projection-based correlation incurs additional cost from computing and maintaining task-specific gradient subspaces, whereas loss-distribution-based correlation relies on lightweight response-level statistics and is therefore substantially more memory efficient. Backward refinement further adds computation proportional to the number of selected prior tasks and the refinement budget. Memory usage is reported as the total memory footprint during training and includes both prompt parameters and any auxiliary statistics maintained by the correlation criterion. To ensure fair comparison and reproducibility, all methods are evaluated with identical training configurations for a given backbone.

## F. Additional Results

**Forward transfer analysis.** Table 14 reports forward transfer (FWT) on T5-LARGE, which measures how learning previous tasks influences performance on newly arriving tasks. Higher FWT indicates that earlier tasks provide beneficial initialization or shared representations for subsequent tasks, while negative values indicate limited or harmful forward influence. Across baselines, methods such as SAPT and SHLPT achieve strong forward transfer by explicitly promoting prompt sharing or reuse across tasks. Notably, SHLPT is explicitly designed to *reduce negative transfer* by learning a task-similarity estimator and adapting the transfer strategy based on whether prior tasks are similar or dissimilar, which aligns closely with the forward-transfer-style metric (where negative values indicate negative transfer) and can therefore yield stronger FWT than methods primarily targeting backward refinement. In contrast, SABER is primarily designed to enable *backward* knowledge transfer through selective refinement of prior prompts, rather than to maximize forward transfer. Consequently, SABER-FPP exhibits near-zero or slightly negative FWT, reflecting its emphasis on task isolation.

*Table 14.* Forward Transfer (FWT↑) on **Long Sequence** and **SuperNI** for T5-Large. Higher values indicate stronger forward transfer to new tasks.

| Method | Long Sequence | | SuperNI | |
|---|---|---|---|---|
| | **Order1** | **Order2** | **Order1** | **Order2** |
| Replay | 0.28 | 0.76 | -1.56 | -1.43 |
| L2P | -2.32 | -1.66 | -19.23 | -19.54 |
| LFPT5 | -1.14 | -0.48 | -0.78 | -0.44 |
| ProgPrompt | -4.14 | -5.58 | -3.43 | -4.98 |
| CODA Prompt | 0.38 | 0.46 | -0.65 | -0.87 |
| SHLPT | 0.62 | 0.67 | 0.32 | **1.98** |
| SAPT | 1.85 | **2.87** | **1.87** | 1.32 |
| **SABER-FPP** | -0.65 | -0.32 | -1.19 | -0.93 |
| **SABER-SPA** | **0.12** | 0.34 | 0.12 | **0.54** |

SABER-SPA achieves modest positive FWT, as shared prompt augmentation allows limited forward reuse while still preserving backward refinement safety. These results highlight a trade-off between forward and backward transfer: while some methods optimize forward transfer at the cost of backward interference, SABER prioritizes stable backward transfer while maintaining competitive forward transfer where shared prompt structures are permitted.

*Table 15.* Memory overhead (KB) of task-correlation strategies as the number of tasks increases. Memory growth is independent of the backbone given fixed prompt dimensionality.

| Number of Tasks | Gradient-based (KB) | Loss-based (KB) |
|---|---|---|
| 5 | 800 | **20** |
| 10 | 1600 | **40** |
| 15 | 2400 | **60** |

## G. Intuition behind loss-distribution based correlation.

Figure 7 provides an intuitive illustration of the proposed loss-based correlation criterion. We visualize the empirical loss distributions of two tasks under a frozen backbone before and after learning a task-specific prompt. In the top panel, the backbone without prompts induces distinct loss distributions $L_1^{(0)}$ and $L_2^{(0)}$ for Tasks 1 and 2, resulting in a large Wasserstein distance $d^{(0)} = W(L_1^{(0)}, L_2^{(0)})$. After training a prompt $u_1$ on Task 1 (bottom panel), the induced loss distributions $L_1^{(1)}$ and $L_2^{(1)}$ become closer, yielding a smaller distance $d^{(1)} = W(L_1^{(1)}, L_2^{(1)})$. The reduction $d^{(0)} - d^{(1)}$ captures the extent to which optimizing the prompt for Task 1 also improves the loss behavior on Task 2, indicating correlation between Task 1 and Task 2. Importantly, this criterion operates at the level of model responses rather than static semantic similarity, allowing it to detect task relationships that emerge through optimization dynamics.

## H. Qualitative visualization of dataset-level representation for long sequence benchmark

Figure 8 provides a qualitative view of dataset-level representation structure under different sentence encoders. Each point corresponds to an individual example, with colors indicating dataset identity. We observe that embeddings produced by sentence-transformers/all-MiniLM-L6-v2 (right) and EmbeddingGemma-300M (left) exhibit clearer clustering and separation across datasets. In particular, semantically distinct datasets such as Yelp, QQP, and WiC form more compact and well-separated clusters under MiniLM, while related datasets (e.g., MNLI, RTE, and CB) show partial overlap consistent with their semantic similarity. This visualization suggests that stronger sentence encoders yield more discriminative dataset-level representations, which can facilitate more reliable task similarity estimation and help explain the effectiveness of representation-based or loss-based correlation signals used in selective backward refinement.

# I. Sacalibility

Figure 9 examines the effect of model scaling on average performance (AP) and backward transfer (BWT) across different backbone sizes, ranging from T5-Large (0.77B) to Qwen-3 (4B) and LLaMA-2 (7B). While all methods benefit from increased model capacity in terms of AP, SABER consistently achieves the highest performance across all backbones. More importantly, SABER is the only method that maintains strong and consistently positive BWT at all scales, indicating effective backward knowledge transfer that does not diminish with larger models. In contrast, ProgPrompt and SHLPT exhibit negative or near-zero BWT regardless of model size, suggesting that scaling alone is insufficient to enable backward transfer.

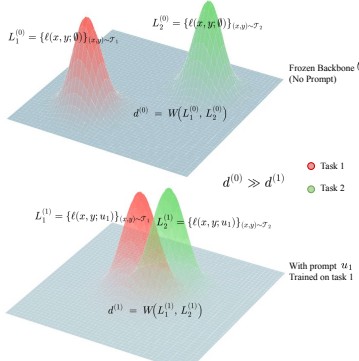

*Figure 7.* Loss-based correlation illustration. The top panel shows the loss distributions of two tasks under a frozen backbone without prompts, where the distributions are well separated and yield a large Wasserstein distance $d^{(0)}$. The bottom panel shows the loss distributions after training a prompt on Task 1, which brings the two distributions closer and reduces the distance to $d^{(1)}$.

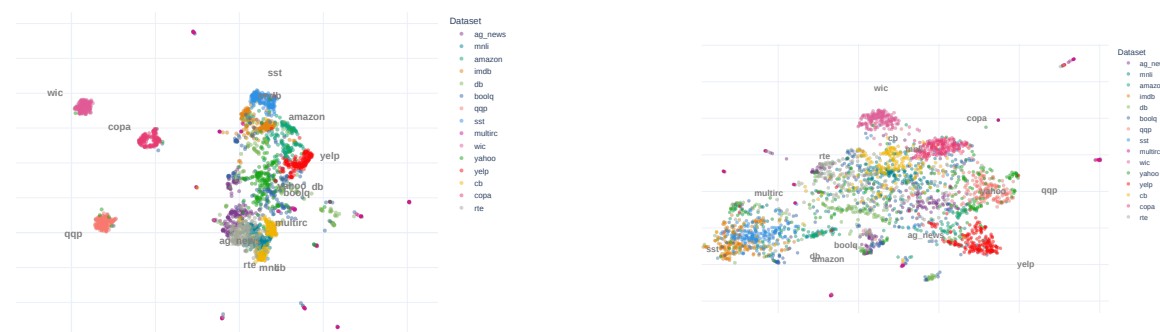

*Figure 8.* Qualitative visualization of dataset-level representations under different encoders. **Left:** embeddings produced by EmbeddingGemma-300M. **Right:** embeddings produced by sentence-transformers/all-MiniLM-L6-v2. Each point corresponds to an example and colors denote datasets.

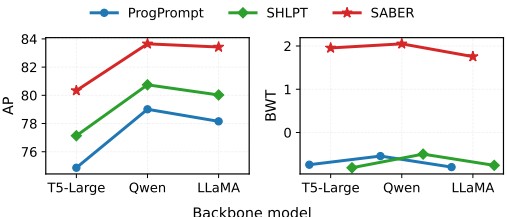

*Figure 9.* Effect of model scaling across T5-Large (0.77B), Qwen-3 (4B), and LLaMA-2 (7B).

*Table 16.* Quantities stored per prior task $i$ in SABER.

| Stored Quantity | Purpose |
|---|---|
| Prompt $u_i$ | Inference and backward refinement |
| Protected subspace $\tilde{U}_i$ | Safe backward refinement |
| Gradient subspace $U_i$ (SABER-P only) | Projection-based task correlation |
| Mean gradient $\bar{g}_i$ (SABER-P only) | Gradient alignment for task correlation |
| WSS score (SABER-L only) | Loss-based task correlation |

## J. Relationship Between Task Similarity and Pairwise BWT

Table 17 reports measured task-correlation scores alongside pairwise BWT for representative task pairs on the Long Sequence benchmark. Task correlation is measured using the projection-based score $s_i$ (Eq. 1). The results show that higher task-correlation scores generally align with stronger pairwise BWT, while low-similarity pairs tend to yield weak or negative backward transfer. Importantly, conceptual or semantic similarity alone is insufficient: pairs that appear related may still fail to produce positive BWT without safe refinement, which is precisely why SABER uses optimization-based criteria, gradient-space compatibility or loss-response alignment, rather than relying on semantic intuition.

*Table 17.* Task-correlation score and pairwise BWT for representative task pairs on the Long Sequence benchmark. Higher similarity generally corresponds to better pairwise BWT.

| Pair | Similarity | BWT |
|---|---|---|
| Yelp $\rightarrow$ Amazon | 0.44 | 1.92 |
| MNLI $\rightarrow$ RTE | 0.35 | 1.41 |
| MultiRC $\rightarrow$ Yahoo | 0.22 | 0.63 |
| RTE $\rightarrow$ WiC | 0.23 | 0.58 |
| BoolQ $\rightarrow$ WiC | 0.14 | 0.06 |
| COPA $\rightarrow$ Yahoo | 0.09 | -0.21 |
| IMDb $\rightarrow$ WiC | 0.07 | -0.34 |
| QQP $\rightarrow$ Amazon | 0.05 | -0.47 |

