# OpenReview forum: "Turning Back Without Forgetting: Selective Backward Refinement for Parameter-Efficient Continual Learning"
_ICML.cc/2026/Conference — ICML 2026 regular_

### Official Review · Reviewer_BppT · 2026-03-08

**Soundness:** 3
**Presentation:** 3
**Significance:** 3
**Originality:** 3
**Overall Recommendation:** 4
**Confidence:** 3

**Summary:**

This paper proposes to improve backward transfer for a prompt-based parameter allocation technique for Continual Learning (CL). The paper first proposes to detect the prompt parameters that are suitable for backward refinement using parameter-level and output-level similarity metrics. Then the paper proposes to selectively update the parameter in the direction of least gradient conflict. Empirical results are provided that provide the motivation behind focusing on selective backward transfer and comparison of the proposed method with CL benchmarks.

**Compliance With Llm Reviewing Policy:**

Affirmed.

**Final Justification:**

My main concerns are mostly addressed in the rebuttal and reply rebuttal. After considering other reviews and corresponding rebuttals, I want to maintain my original score.

Reason for not giving a higher score: This method is limited to prompt-based settings. In the rebuttal, the authors show their method is not easily transferable to a LoRA-based setting. Also, the AP score for prompt-based settings is significantly lower than that of the LoRA-based setting. This limits the overall impact of this work. However, as a prompt-based setting is computationally cheaper by a significant margin, this method/paper is still significant.

**Key Questions For Authors:**

Please refer to the weakness section.

**Strengths And Weaknesses:**

**Strengths:**
- Selective backward refinement is an interesting and important topic for CL.
- The proposed method to detect candidate prompt parameters and principled refinement of selected prompt parameters is novel.
- Empirical results and ablation studies show improvement in both final accuracy and backward transfer.
- Theoretical guarantees about the monotonic improvement of the current task loss during refinement and about refinement directions being orthogonal to protected directions are provided.
- Overall, a well-written paper with a novel idea.

**Weaknesses:**
- Although a discussion about the previous method that improves backward transfer is provided, the empirical comparison with such methods is missing. For instance, the paper’s method can be adapted to LoRA-based parameter allocation or the method from [1] can be adapted for prompts and compared with the paper's setting.
  - [1]: “Turning the Tables: Enabling Backward Transfer via Causal-Aware LoRA in Continual Learning” - NeurIPS 2025
- Prompt-based methods are important because they are parameter-efficient. However, the proposed method can be easily applied to LoRA-based or Adapter-based parameter allocation. Empirical results of using the proposed method for LoRA-based or Adapter-based parameter allocation would further strengthen the paper.
- Although the discussion about the selection of hyperparameters $\tau_s$ and $\tau_{WSS}$ is provided in the Appendix, the experimental results are missing.
- Other related work that needs to be discussed and possibly compared with:
  - Wu et al., “Exploiting Task Relationships in Continual Learning via Transferability-Aware Task Embeddings” - NeurIPS 2025

---

> ### Author Rebuttal · Authors · 2026-03-30
>
> We thank the reviewer for the valuable suggestions and thoughtful feedback.
>
> **Q1**
>
> A direct comparison to [1] is not feasible because the method is built around LoRA-specific parameters and updates. In [1], each task is represented by two trainable low-rank matrices​ and​ inserted inside the model’s attention layers, and both the update rule and backward-transfer mechanism are defined directly on these matrices and their gradients. Their PaCA module estimates the causal effect of individual entries of the LoRA weight matrix, and CaGA module computes cross-task correlation and affinity using LoRA-layer gradients.
>
> In prompt-based CL, these objects simply do not exist in the same form: there are no inserted low-rank adapter matrices inside backbone layers, and the learnable variables are instead soft prompts which sit outside the model. Extending [1] to prompts would require redefining the parameter space, the causal-effect computation, the task-correlation measure, and the refinement rule.
>
> We will clarify this in the revision
>
>
> ---
>
>
>
> **Q2**
>
>  We agree that extending SABER to LoRA or adapters is an interesting direction. To test whether this extension is immediate, we implemented a straightforward LoRA version by directly treating task-specific LoRA parameters as the counterpart of  task-specific prompts, compute task correlation in that parameter space, and apply constrained backward refinement to non-conflicting directions of prior task-specific updates. However, this naive extension did not yield the same gains as SABER in prompt space, and in our preliminary experiments it led to weak or even negative BWT (Table Below).
>
> | Method | Order 1 (Long Sequence) |
> | --- | --- |
> | LoRA baseline | AP - 79.8 &nbsp;&nbsp; BWT - -1.7 |
> | + naive SABER extension | AP - 79.4 &nbsp;&nbsp; BWT - -0.9 |
>
> We believe this happens for two main reasons:
>
> (1) **The refinement space is fundamentally different**: For LoRA, each task is represented by two coupled low-rank matrices (A and B) rather than a single prompt parameter, so even defining the protected subspace is non-trivial: one must decide whether to protect the parameters themselves (A and B) or the composed update (BA) they induce in the backbone.
>
> (2) **Safe backward refinement is less direct**. For prompts, the update is applied to a single prompt parameter, so it is clear how to project it to avoid interference. For LORA, the trainable parameters are part of an internal module, so the notion of a safe non-interfering update is no longer immediate. This means the correlation measure and refinement rule must be redesigned for that module.
>
> Overall, these results suggest that extending SABER beyond prompts is possible, but not a drop-in extension. It requires rethinking the refinement space, the task-correlation measure, and the non-interference constraint for the target PEFT module. We will clarify this in the revision and highlight LoRA/adapter extensions as important future work.
>
> ---
>
> **Q3**
>
> While the appendix discusses the threshold choices, we agree that including the corresponding sensitivity results would make the robustness claim more concrete, and we provide them below.
>
> **Sensitivity Analysis of $\tau_s$ and $\tau_{WSS}$ (T5-Large, Long Sequence Order 1):**
>
> **Saber-P**
>
> | $\tau_S$ | 0.01 | 0.05 | 0.10 | 0.20 | 0.40 | 0.60 |
> |:--------:|:----:|:----:|:----:|:----:|:----:|:----:|
> | AP       | 78.1 | 79.8 | **80.3** | 80.1 | 78.9 | 77.2 |
> | BWT      | -0.8 | 1.2  | **2.1**  | 1.9  | 1.2  | 0.4  |
>
> **Saber-L**
>
> | $\tau_{\mathrm{WSS}}$ | 0.05 | 0.10 | 0.20 | 0.35 | 0.50 | 0.70 |
> |:---------------------:|:----:|:----:|:----:|:----:|:----:|:----:|
> | AP                    | 77.5 | 79.2 | **80.3** | 80.0 | 79.1 | 77.8 |
> | BWT                   | -1.1 | 1.0  | **1.9**  | 1.8  | 1.1  | 0.3  |
>
>
> Very small thresholds select too many prior tasks for refinement, increasing negative transfer and hurting both AP and BWT. Very large thresholds are overly conservative, selecting very few prior tasks and thus reducing backward-transfer gains. The best results occur in a moderate range around the default values $(\tau_s = 0.1, \tau_{\mathrm{WSS}} = 0.2)$, indicating that SABER is reasonably robust and does not require fine-grained threshold tuning.
>
> ---
>
> **Q4**
>
> [1] propose to use task relationships to improve continual learning, but they do so through transferability-aware task embeddings that guide a hypernetwork to generate task-conditioned parameters. In contrast, our method focuses on prompt-based continual learning and addresses a different question: how to enable positive backward transfer by deciding which prior prompts should be refined and how to refine them safely without interfering with previously learned knowledge. So, the two works are related in that both use task relationships, but they differ in both mechanism and setting.
>
> We will clarify this in the related work and add comparison results in the revision.

---

> > ### Author Rebuttal · Reviewer_BppT · 2026-04-03
> >
> > Thank you for the rebuttal. However, I do not think the response fully addresses the comparison request.  The paper still needs to justify not empirically considering the methods that improve backward transfer. Additionally, [1] is also a PEFT-based CL method, which compares with prompt-based methods. Also, "prompts sit outside the model" is not accurate, as past methods also inject prompts in intermediate layers. Overall, the response to Q1, Q2, and Q4 does not sound convincing.

---

> > > ### Author Response · Authors · 2026-04-03
> > >
> > > We thank the reviewer for the clarification. We agree that, in the LoRA paper [1], a direct comparison across PEFT methods is given but that one is without adapting the method itself to prompt space, and we can also add such a table below. In that comparison, SABER shows stronger BWT, while LoRA-based methods have higher AP, which is expected given their larger trainable capacity (about 10K parameters per task in our T5-Large setup versus roughly 1.6M–2.4M per task for [1]). Our main point is that method-level adaptation is non-trivial in either direction: extending SABER to LoRA requires redesigning the refinement space, correlation measure, and safe update rule for coupled low-rank matrices, while adapting [1] to prompt-based CL is likewise not immediate, especially without a public codebase.
> > >
> > > [1]: “Turning the Tables: Enabling Backward Transfer via Causal-Aware LoRA in Continual Learning” - NeurIPS 2025.
> > >
> > > **Comparison of CaLoRA and SABER on Long Sequence Benchmark**
> > >
> > > | Method      | Order1                  | Order2                   |
> > > |-------------|-------------------------|--------------------------|
> > > | CaLora [1]  | AP: 84.62 BWT: 0.29    |    AP: 81.95 BWT: -0.57    |
> > > | SABER (Ours)| AP: 80.12 BWT: 2.13    | AP: 79.94 BWT: 3.12     |
> > >
> > > Thank you for pointing this out. We agree that this wording was inaccurate, since prompts can also be injected into intermediate layers. Our intended point was narrower: in our method, the task-specific prompts are implemented as external prompt parameters, rather than LoRA-style low-rank weight updates inside the backbone. Because of this difference in parameterization, a direct method-level adaptation is not immediate.

---

### Official Review · Reviewer_eXkK · 2026-03-10

**Soundness:** 2
**Presentation:** 2
**Significance:** 2
**Originality:** 3
**Overall Recommendation:** 4
**Confidence:** 4

**Summary:**

The paper proposes SABER, a replay-free framework for enabling backward knowledge transfer in prompt-based continual learning. SABER addresses limited backwards transfer as a result of freezing by introducing two task correlation criteria, a projection that measures gradient subspace alignment between tasks and a loss distribution to compare model similarity. The authors provide theoretical guarantees that these updates do not destructively interfere and that the resulting loss does not increase under mild smoothness conditions.

**Compliance With Llm Reviewing Policy:**

Affirmed.

**Final Justification:**

The rebuttal addressed many of my concerns. I am increasing my score.

**Key Questions For Authors:**

- How sensitive is SABER to the choice of subspace rank $r$?
- In Table 7, removing task correlation causes BWT to collapse far more severely than removing constrained refinement alone. Does this suggest that task selection is the more critical component, and could a simpler unconstrained update suffice when selection is highly accurate?
- Do the authors have results showing similarity between tasks, and how this correlates to the task-specific accuracy? It would be interesting to see if BWT is better between conceptually similar tasks.

**Limitations:**

Yes

**Strengths And Weaknesses:**

**Strengths**
- The paper is well written and presented.
- Theoretical guarantees are appreciated and back up empirically verified claims.
- The modular design of the method strengthens practical applicability, as it can be adapted to other continual learning pipelines.
- Experiments span multiple task orderings and backbones. Strong backwards transfer results are reported.

**Weaknesses**
- The projection-based variant introduces non-trivial computational and memory overhead, requiring per-task gradient subspaces maintained via SVD, which could become prohibitive in very long task sequences or high-dimensional prompt settings. A further theoretical discussion of the computational complexity could be provided, and while wall clock timings and memory results are provided, there is limited comparison to SOTA methods.
- While broad benchmark datasets are used, results over Long Sequence and SuperNI are often reported as overall averages. The specific generalisability of the method is obscured in this way. Moreover, task-specific accuracies in Figure 5 shows significant variability in performance across training. There is limited analysis into why this is the case.
- Evaluation is restricted to task-incremental settings with known task identities at inference, which limits generalisability to more realistic class-incremental or domain-incremental scenarios that are common in practical deployments.

---

> ### Author Rebuttal · Authors · 2026-03-30
>
> We thank the reviewer for the thoughtful feedback.
>
> **W1:**
> We agree that the projection-based variant introduces additional overhead, since it requires maintaining per-task gradient subspaces. This is precisely why we also propose loss-based correlation variant, which avoids storing gradient subspaces for task selection and is more scalable option for long task sequences. That said, we also acknowledge that safe backward refinement still requires storing the cumulative protected subspace for previously refined prompts, so some memory overhead remains intrinsic to preserving non-interference.
>
> In terms of complexity, the main extra cost in SABER comes from (i) task-correlation estimation and (ii) safe backward refinement of selected prior prompts. Let $\ell d$ denote prompt dimension, $r$ protected-subspace rank, $K$ number of refinement updates, and $|S_t|$ number of selected prior prompts. Both SABER-P and SABER-L incur refinement cost on the order of $O(|S_t| K \ell d r)$ due to protected-subspace projection during backward updates. SABER-P has additional cost for gradient-subspace-based correlation estimation, while SABER-L replaces this selection stage with Wasserstein-based loss-distribution.
>
> We report overhead relative to ProgPrompt (please check Table in weakness 2 response to reviewer mEqe), since it is one of the most efficient prompt-based SOTA baselines and thus a strong reference point for isolating SABER’s cost. We expect similar trade-offs relative to other SOTA methods as well, and we will include them in the revision.
>
> ---
>
> **W2:**
> Beyond overall averages, we have also included task-level analyses: Fig. 3 shows per-task gains from backward refinement, and Fig. 5 reports task-wise final accuracies with and without backward refinement. These results show benefits across tasks and settings.
>
> The variability in Fig. 5 is expected in heterogeneous benchmarks such as Long Sequence and SuperNI, where tasks differ substantially in difficulty and transferability. The key point is that, despite this natural variability, SABER consistently shifts task outcomes upward relative to the no-BWT baseline and maintains positive BWT.
>
> ---
>
> **W3:**
> We agree that extending to class-incremental or domain-incremental settings is an important direction. However, our paper is explicitly scoped to task-incremental continual learning, which is also the standard setting used by recent papers [1]–[3]. Extending SABER beyond the task-incremental setting is nontrivial, and an important future direction.
>
> [1] Ca-Lora (NeurIPS 2025)
> [2] SHLPT (ACL 2024)
> [3] SAPT (ACL 2024)
>
> ---
>
> **Q1:**
> SABER is not overly sensitive to the subspace rank, when $r$ is small. We use $r=3$ consistently across all experiments, which we found to provide a good balance between protecting prior-task directions and preserving room for beneficial backward refinement, while keeping the overhead low. If $r$ is too small, the protected subspace may miss important directions; if it is too large, refinement becomes overly conservative and more expensive. We also experimented with a larger rank ($r=5$) and found that performance was generally comparable, and in some cases slightly worse, while incurring higher memory overhead; supporting $r=3$ as the default.
>
> ---
>
> **Q2:**
> If backward updates are applied to all previous tasks, performance deteriorates on unrelated tasks. Table 7 suggests that task correlation is the first safeguard, since it prevents harmful refinement on prior tasks that are unlikely to benefit from transfer. However, selection alone is not always sufficient: as shown in Table 3, even after a prior task is selected, unconstrained updates are generally less reliable than the orthogonally constrained update because they can still interfere with task-critical directions. In this sense, task selection and constrained refinement play complementary roles: selection identifies where to refine; constraints keep it safe.
>
> ---
>
> **Q3:**
> Task correlation is generally associated with better pairwise BWT, but conceptual similarity alone is not sufficient. In the paper, we already show that unrelated pairs consistently yield negative pairwise BWT under unconstrained backward updates, while related pairs are more promising but can still fail without safe refinement (Table 1, Table 2). This is precisely why SABER uses task-correlation criteria based on optimization behavior, either gradient-space compatibility or loss-response alignment, rather than relying only on semantic intuition.
>
> | Pair | Similarity | BWT |
> | --- | ---: | ---: |
> | Yelp → Amazon | 0.44 | 1.92 |
> | MNLI → RTE | 0.35 | 1.41 |
> | MultiRC → Yahoo | 0.22 | 0.63 |
> | RTE → WiC | 0.23 | 0.58 |
> | BoolQ → WiC | 0.14 | 0.06 |
> | COPA → Yahoo | 0.09 | -0.21 |
> | IMDb → WiC | 0.07 | -0.34 |
> | QQP → Amazon | 0.05 | -0.47 |
>
> The table shows that higher measured similarity generally aligns with better pairwise BWT, while low-similarity pairs are often weak or negative, supporting our claim.

---

> > ### Author Rebuttal · Reviewer_eXkK · 2026-04-01
> >
> > My concerns are well addressed. I am willing to increase my overall score from 3 to 4.

---

> > > ### Author Response · Authors · 2026-04-03
> > >
> > > We thank the reviewer for taking the time to read our rebuttal and for raising the score. We truly appreciate constructive feedback throughout this process.

---

### Official Review · Reviewer_7eTe · 2026-03-12

**Soundness:** 3
**Presentation:** 3
**Significance:** 3
**Originality:** 2
**Overall Recommendation:** 5
**Confidence:** 4

**Summary:**

This manuscript considered the prompt-based continual learning with pre-trained models (CL-PTM). Existing relevant studies usually apply the knowledge isolation to the task-specific prompts, but this isolation also impedes the potential backward knowledge transfer (BWT) that aims to improve the performance on previous tasks after learning on the successive tasks. Based on this consideration, this manuscript focuses on improving the BWT with soft prompt refinement. With the empirical analysis, the authors pointed out that 1) the backward refinement must be only triggered on previous tasks that are compatible with the learning signals in the current task; 2) even for compatible tasks, the BWT can be negligible or even negative if the backward refinement is unconstrained. Then, the authors proposed the framework of Selective Backward Refinement for Positive Backward Knowledge Transfer (SABER), which can determine when and how to apply the backward refinement. Experiments on some recent large language models (LLMs) are conducted to support the proposed framework.

**Compliance With Llm Reviewing Policy:**

Affirmed.

**Final Justification:**

I appreciated the further explanations from the authors during the rebuttal process. My concerns have been addressed.
Based on this, I will maintain my current score and hold my current position (acceptance) to this manuscript.

**Key Questions For Authors:**

1. In Figure 4, the authors provided some statistics regarding the scalability. Could the authors explain more details about how they compute the memory overhead within this figure?
2. In the task similarity measurement and backward refinement, it seems that we need to store some information from the previous tasks, including but not limited to the gradient information. Could the authors provide a checklist about which matrices we should save during the training process?
3. The experiments are conducted on task sequences of length 15. Could the authors comment on how the proposed method would scale to substantially longer task sequences? In particular, how would the protected subspace and stored statistics grow with the number of tasks?

**Strengths And Weaknesses:**

## Strengths
1. This work provided a novel perspective on positive backward knowledge transfer in continual learning, which is usually neglected in previous studies that only considered reducing forgetting.
2. The proposed framework is straightforward: determine which previous tasks should be refined (with certain task similarity measures) and how these tasks should be refined (by applying the refinement constraints). The thread of this manuscript is clear to readers.
3. The authors tried to provide theoretical analyses to support their claims.
4. This work considered the more practical experimental settings. For example, the experiments were conducted on some practical LLM models such as LLaMa and QWen, which can provide more practical insights compared to previous studies that only conducted experiments on smaller models such as ViT.

## Weaknesses
1. The proposed method is designed specifically for prompt-based continual learning. Some techniques, like gradient projection for prompt update is not novel. I didn't see a significant new message regarding the constraints on prompt refinement.
2. Although Figure 4 discussed a little about the scalability, it seems that the discussion of the computational and memory overhead was still limited. It seems that what Figure 4 accounted for and how the authors assessed the computational and memory head were still not that clear.
3. Although the authors claimed the robustness when choosing the thresholds for task similarity, I still wonder if it would be difficult to determine them for new tasks/models/datasets.

---

> ### Author Rebuttal · Authors · 2026-03-30
>
> We thank the reviewer for the valuable suggestions and thoughtful feedback.
>
> **Weakness 1**
>
> We agree that gradient projection itself is not new. The novelty of SABER lies in why and how it is used: not as a generic constrained update, but as a mechanism for backward prompt refinement in prompt-based continual learning. In particular, we use projection to construct a nontrivial cumulative protected subspace, which allows prior prompts to be refined only in non-conflicting directions across multiple future tasks. The key message behind the constraint: together with task selection, it makes backward transfer possible without damaging prior knowledge, and it also comes with a theoretical performance guarantee.
>
> ---
>
> **Weakness 2**
>
> We thank the reviewer for this important question. In our experiments, the same global thresholds remained stable across a wide range of settings, including different task sequences (Long Sequence and SuperNI; Table 8), multiple task orders (Order 1 and Order 2; Table 9), and multiple backbone models (T5-Large, LLaMA, and Qwen). We also tested a broader range of threshold values and observed similar task selections and comparable performance within a moderate range. For a completely new dataset or backbone model, some threshold tuning may still be helpful. However, given the diversity of our current experiments, we believe the threshold choice is reasonably robust in practice.
>
>
> ---
> **Weakness 3 and Question 1**
>
> We thank the reviewer for this question. Figure 4 reports the memory overhead of the two task-correlation methods. The two SABER variants store different per-task statistics for correlation estimation.
>
> For projection-based correlation, for each prior task we store: (i) the mean gradient direction $\bar{g}_i \in \mathbb{R}^{\ell d}$, and (ii) a rank-$r$ gradient subspace basis $U_i \in \mathbb{R}^{(\ell d) \times r}$. Thus, the per-task storage is $\ell d + \ell d r = \ell d (r+1)$ float values. With our setting $\ell = 10$, $d = 1024$, and $r = 3$, this corresponds to about $160$ KB per task.
>
> For loss-based correlation, we store only lightweight scalar loss statistics used for Wasserstein-based similarity, which requires about $4$ KB per task in our implementation. This matches Figure 4, which reports $800/1600/2400$ KB for the gradient-based strategy and $20/40/60$ KB for the loss-based strategy at $5/10/15$ tasks, respectively.
>
> ---
> **Question 2**
>
> We thank the reviewer for this question. The quantities that need to be stored from previous tasks are summarized below:
> | Stored quantity from prior task $i$ | Purpose |
> | --- | --- |
> | Prompt $u_i$ | Inference and backward refinement |
> | Protected subspace $\tilde{U}_i$ | Safe backward refinement |
> | Gradient subspace $U_i$ (SABER-P only) | Projection-based task correlation |
> | Mean gradient $\bar{g}_i$ (SABER-P only) | Gradient alignment for task correlation |
> | Loss-based statistic / WSS score (SABER-L only) | Loss-based task correlation |
>
> ---
> **Question 3**
>
> We thank the reviewer for this important question. Beyond 15 tasks, the task-specific prompts in SABER grow linearly with the number of tasks, as in standard prompt-based continual learning. In addition, both SABER-P and SABER-L maintain a protected subspace for each learned prompt, so this component also grows linearly with the number of tasks. The key difference is in the task-correlation statistics: SABER-P stores gradient-based information for each prior task, whereas SABER-L stores only lightweight loss-based statistics. As a result, both variants grow nearly linearly with sequence length, SABER-L uses much less memory and scales better to long task sequences, whereas SABER-P uses more memory in exchange for more detailed gradient information. We will clarify this distinction explicitly in the revision.

---

> > ### Author Rebuttal · Reviewer_7eTe · 2026-04-03
> >
> > My questions have been addressed during the rebuttal phase.

---

> > > ### Author Response · Authors · 2026-04-05
> > >
> > > We thank the reviewer for taking the time to read our rebuttal. We truly appreciate constructive feedback throughout this process.

---

### Official Review · Reviewer_mEqe · 2026-03-13

**Soundness:** 3
**Presentation:** 3
**Significance:** 2
**Originality:** 3
**Overall Recommendation:** 4
**Confidence:** 2

**Summary:**

This paper addresses the knowledge backward transfer problem in LLM continual learning framework, and propose a prompt based method SABER to solve it. SABER selectively refiens previously learned parameters upon new tasks' arrival, utilizing a complementary task-correlation critria and orthogonal updating strategy. Theoretical results show that the refinements steps does not affect the protected direction and curren task loss. Empirical validations on several language models further demonstrate the consistent gains in average CL performance and backward trasnfer.

**Compliance With Llm Reviewing Policy:**

Affirmed.

**Final Justification:**

The authors have well addressed my concerns and I will maintain my recommendation as weak accept.

**Key Questions For Authors:**

Please see the weakness part.

**Limitations:**

yes.

**Strengths And Weaknesses:**

**Strength**:

This paper presents a well-motivated, replay-free framework that enables positive backward transfer in prompt-based continual learning by safely updating previously frozen prompts. The methodology is technically innovative, utilizing complementary task-correlation criteria alongside orthogonal updates to a protected gradient subspace to prevent catastrophic interference. Supported by a clear presentation and comprehensive ablation studies , the approach is rigorously validated across diverse model architectures, consistently demonstrating stable backward transfer where baseline methods struggle. Overall, this model-agnostic strategy represents a significant and practically viable advancement for PEFT based CL problems.

------

**Weakness**:

1. It would be helpful if the author could provide more details on some of the hyper parameters (seems missing from the manuscript's table 10), such as the subspace rank, the number of refinement steps per prompt, and the learning rate for backward refinement, etc. Also, what's the default buffer size of replay-based methods?

2. Apart from memory usage, could the author provide more details regarding runtime and computational complexity?

3. (Minor issue) Despite the statement in Appendix D about the selection of threshold hyperparameters, can the authors provide a more detailed analysis of $\tau_s$ and $\tau_{wss}$ in a broader range?

---

> ### Author Rebuttal · Authors · 2026-03-30
>
> We thank the reviewer for the valuable suggestions and thoughtful feedback.
>
> **Weakness 1**
>
> We appreciate reviewer for this careful observation. We provide the missing hyperparameter details below and will incorporate them into the revised manuscript.
>
> **Subspace rank r**: We use r=3 across all experiments. We also tried larger ranks (e.g., r=5), but observed comparable or slightly worse performance together with higher memory overhead, so we kept r=3 as the default.
>
> **Number of backward refinement steps**: Backward refinement is performed during the final 2 epochs of current-task training, after the current-task prompt has sufficiently stabilized.
>
> **Learning rate for backward refinement**: We use a smaller learning rate for backward refinement than for forward prompt training - specifically η_backward = 0.001 for both autoregressive models and for sequence-to-sequence models.
>
> **Replay buffer size**: For the Replay baseline, we follow the standard setting from prior work, using 100 samples per task.
>
>
> ---
> **Weakness 2**
>
> We thank the reviewer for this suggestion.  We report total wall-clock training time over the full task sequence, together with the additional overhead relative to the standard prompt-based baseline in the table below. All experiments are run on a single NVIDIA A100 80GB GPU.
>
> | BACKBONE | Method  | Total GPU Time (h:m) | Overhead |
> |:--------:|:-------:|:--------------------:|:--------:|
> | T5-Large | SABER-P | 24:09                | +06:12   |
> | T5-Large | SABER-L | 26:00                | +08:03   |
> | LLaMA    | SABER-P | 105:04               | +32:00   |
> | LLaMA    | SABER-L | 100:08               | +27:04   |
>
> The runtime results indicate that both SABER variants add only moderate overhead. While the exact runtime trade-off varies slightly by backbone, SABER-L remains competitive in runtime and is substantially more memory-efficient, whereas SABER-P tends to be more computationally demanding due to gradient-subspace-based correlation estimation and projection operations.
>
> In terms of complexity, the main extra cost in SABER comes from (i) task-correlation estimation and (ii) safe backward refinement of selected prior prompts. Let $ℓd$ denote the prompt dimension, $r$ the protected-subspace rank, $K$ the number of refinement updates, and $|S_t|$ the number of selected prior prompts. Both SABER-P and SABER-L incur refinement cost on the order of $O(|S_t| K \ell d r)$ due to protected-subspace projection during backward updates. SABER-P has additional cost for gradient-subspace-based correlation estimation, while SABER-L replaces this selection stage with marginal Wasserstein-based loss-distribution comparison.
>
> We will add this runtime table and a short complexity discussion to the revised manuscript for completeness.
>
> ---
>
> **Weakness 3**
>
> We thank the reviewer for this important question. Appendix D currently reports robustness within $\tau_s \in [0.05, 0.2]$ and $\tau_{WSS} \in [0.15, 0.3]$. We agree that a broader analysis is warranted and will add the following ablation table to the revised manuscript.
>
> **Sensitivity Analysis of $\tau_s$ and $\tau_{WSS}$ (T5-Large, Long Sequence Order 1):**
>
> **Saber-P**
>
> | $\tau_S$ | 0.01 | 0.05 | 0.10 | 0.20 | 0.40 | 0.60 |
> |:--------:|:----:|:----:|:----:|:----:|:----:|:----:|
> | AP       | 78.1 | 79.8 | **80.3** | 80.1 | 78.9 | 77.2 |
> | BWT      | -0.8 | 1.2  | **2.1**  | 1.9  | 1.2  | 0.4  |
>
> **Saber-L**
>
> | $\tau_{\mathrm{WSS}}$ | 0.05 | 0.10 | 0.20 | 0.35 | 0.50 | 0.70 |
> |:---------------------:|:----:|:----:|:----:|:----:|:----:|:----:|
> | AP                    | 77.5 | 79.2 | **80.3** | 80.0 | 79.1 | 77.8 |
> | BWT                   | -1.1 | 1.0  | **1.9**  | 1.8  | 1.1  | 0.3  |
>
>
>
> **Interpretation**  Very small thresholds select too many prior tasks for refinement, increasing negative transfer and hurting both AP (Average Performance) and BWT. Very large thresholds are overly conservative, selecting very few prior tasks and thus reducing backward-transfer gains. The best results occur in a moderate range around the default values $(\tau_s = 0.1, \tau_{\mathrm{WSS}} = 0.2)$, indicating that SABER is reasonably robust and does not require fine-grained threshold tuning.

---

> > ### Author Rebuttal · Reviewer_mEqe · 2026-04-04
> >
> > Thanks for the rebuttal efforts. The authors have provided clear clarification and supportive results. All my concerns have been well addressed. Therefore, I maintain my positive recommendation as weak accept.

---

> > > ### Author Response · Authors · 2026-04-05
> > >
> > > We thank the reviewer for taking the time to read our rebuttal. We truly appreciate constructive feedback throughout this process.

---

### Decision · Program_Chairs · 2026-04-30

**Decision:**

Accept (regular)

**Comment:**

This paper focuses on prompt-based continual learning and how backwards transfer can be enabled in such a setting.
All reviewers are favorable regarding the soundness and acceptance of the work.
Whereas some initial requests were made regarding better clarification of e.g. hyper-parameters and specification of compute/memory overheads, these could be fully addressed during the rebuttal period. This has lead to one reviewer raising the score, and another retaining a weak accept score with a comment that concerns have been fully resolved. The only remaining point of possible criticism is a limitation in scope due to narrowing the contribution down to prompt-only continual learning and seemingly being incompatible with some other continual learning techniques.

Overall, the paper thus presents a solid contribution to ICML and the AC recommends to accept the paper.